# Global 7-km mesh nonhydrostatic Model Intercomparison Project for improving TYphoon forecast (TYMIP-G7): Experimental design and preliminary results

Masuo Nakano[1], Akiyoshi Wada[2], Masahiro Sawada[2], Hiromasa Yoshimura[2], Ryo Onishi[1], Shintaro Kawahara[1], Wataru Sasaki[1], Tomoe Nasuno[1], Munehiko Yamaguchi[2], Takeshi Iriguchi[2], Masato Sugi[2] and Yoshiaki Takeuchi[2]

[1]Japan Agency for Marine-Earth Science and Technology, 3173-25 Showa-machi, Kanazawa-ku, Yokohama, Kanagawa 236-0001, Japan
[2]Meteorological Research Institute, Japan Meteorological Agency, 1-1 Nagamine, Tsukuba, Ibaraki 305-0052, Japan

*Correspondence to*: Masuo Nakano (masuo@jamstec.go.jp)

**Abstract.** Recent advances in high-performance computers facilitate operational numerical weather prediction by global hydrostatic atmospheric models with horizontal resolutions of ~10 km. Given further advances in such computers and the fact that the hydrostatic balance approximation becomes invalid for spatial scales <10 km, the development of global nonhydrostatic models with high accuracy is urgently required.

The Global 7-km mesh nonhydrostatic Model Intercomparison Project for improving TYphoon forecast (TYMIP-G7) is designed to understand and statistically quantify the advantages of high-resolution nonhydrostatic global atmospheric models to improve tropical cyclone (TC) prediction. A total of 137 sets of 5-day simulations using three next-generation nonhydrostatic global models with horizontal resolutions of 7 km and a conventional hydrostatic global model with a horizontal resolution of 20 km were run on the Earth Simulator. The three 7-km mesh nonhydrostatic models are the nonhydrostatic global spectral atmospheric Model using Double Fourier Series (DFSM), the Multi-Scale Simulator for the Geoenvironment (MSSG), and the Nonhydrostatic ICosahedral Atmospheric Model (NICAM). The 20-km mesh hydrostatic model is the operational Global Spectral Model (GSM) of the Japan Meteorological Agency.

Compared with the 20-km mesh GSM, the 7-km mesh models reduce systematic errors in the TC track, intensity and wind radii predictions. The benefits of the multi-model ensemble method were confirmed for the 7-km mesh nonhydrostatic global models. While the three 7-km mesh models reproduce the typical axisymmetric mean inner-core structure, including the primary and secondary circulations, the simulated TC structures and their intensities in each case are very different for each model. In addition, the simulated track is not consistently better than that of the 20-km mesh GSM. These results suggest that the development of more sophisticated initialization techniques and model physics is needed to further improve the TC prediction.

## 1 Introduction

### 1.1 Global model

Global models provide fundamental information for operational weather forecasting at daily, weekly and seasonal time scales. Moreover, such models furnish initial and lateral boundary conditions to limited-area models, which furnish fundamental information for local-scale weather forecasts. Therefore, operational numerical weather prediction centres have been developing sophisticated global models with high resolution and accuracy. Because such models require huge computational resources, their development strongly depends on advances in high-performance computers. Recent computer progress has facilitated the reasonable operation of global models with horizontal resolutions of ~10 km. Indeed, the European Centre for Medium-Range Weather Forecasts (ECMWF) has operated a global model with a horizontal resolution

of 9 km since March 2016. Therefore, sooner or later, it is expected that all numerical weather prediction centres will operate global models with horizontal grid intervals of <10 km.

Developing high-resolution models with a horizontal grid spacing of <10 km must resolve three challenges. The first is to use a nonhydrostatic equation system. In the Earth's atmosphere, hydrostatic balance is established for spatial scales >10 km with high accuracy. Therefore, the primitive equation system, which approximates the vertical momentum equation with the hydrostatic balance equation, has been used in conventional global models. The second challenge is to use a dynamical core that effectively runs on state-of-the-art, massively parallel computer systems. Many conventional global models use the spectral method in which the Legendre transform is used for the meridional expansion of certain prognostic variables. Because the computational cost of this transform increases with the third power of the number of grid points and communication costs become large, one solution is to avoid such transforms (Tomita et al., 2001). The last challenge is to implement sophisticated physical schemes suitable for high-resolution models, especially for clouds, because they can be partially resolved in a model with a horizontal resolution of 10 km.

Because developing operational numerical weather prediction models with high accuracy requires huge computational and human resources, the concept of transition of research to operations (R2O) has recently been encouraged. For example, the Hurricane Weather Research and Forecasting Model (Bernardet et al., 2015) and an atmosphere–ocean coupled limited-area model (Ito et al., 2015) have been developed based on R2O in the United States and Japan, respectively. In Japan, two next-generation, nonhydrostatic global atmospheric models have already been developed and used in the research community. These are called the Multi-Scale Simulator for the Geoenvironment (MSSG) and the Nonhydrostatic ICosahedral Atmospheric Model (NICAM). In addition, the Meteorological Research Institute (MRI) of the Japan Meteorological Agency (JMA) has developed a next-generation nonhydrostatic atmospheric model called the nonhydrostatic global spectral atmospheric Model using a Double Fourier Series (DFSM). To gain knowledge, to develop and improve nonhydrostatic global models and to share them with the research and operational communities are some aims of the present project.

## 1.2 TC forecasts

Tropical cyclones (TCs) are characterized by violent winds and torrential rain. These events cause tremendous damage to human lives, property and socioeconomic activity via landslides, floods and storm surges. Because an average of 26 TCs (>30% of the global average) form in the western North Pacific each year, accurate TC track and intensity forecasts are of great concern to East Asian countries to mitigate the impacts of the associated disasters. The JMA has the primary responsibility for TC forecasts in the western North Pacific region as a Regional Specialized Meteorological Centre (RSMC) of the World Meteorological Organization. The JMA has operated a 20-km mesh global atmospheric model to predict weather and TC tracks and intensities since 2007. Therefore, upgrading their global atmospheric model is a promising approach to improve TC forecasts in the western North Pacific.

Errors in track prediction by the JMA operational global atmospheric model have decreased on an average by half over the past 20 years (JMA, 2014) as the operational model has been upgraded. For example, TC track prediction error in a 30-hour forecast with a 60-km mesh global model was ~200 km in 1997 and decreased to ~100 km in 2010 with a 20-km mesh model. Even though we have continuously improved TC track prediction, abnormally large track prediction errors called 'forecast busts' (e.g. Carr and Elsberry, 2000) still occur. Typhoons Conson (2004) (Yamaguchi et al., 2009) and Fengshen (2008) (Yamada et al., 2016; Nasuno et al., 2016) are typical examples. Tracks predicted by tens-of-km mesh global models for Fengshen showed serious poleward-bias recurving far from the Philippine Islands; however, the typhoon made landfall in the Philippines according to best-track analyses (Joint Typhoon Warning Center, 2008). Yamada et al. (2016) reported that a 3.5-km mesh next-generation nonhydrostatic global model successfully simulated its landfall in the

Philippines. Increases in the horizontal resolution of global atmospheric models with appropriate physical schemes can potentially reduce bust cases and annual mean errors of TC track predictions.

Despite the advances in TC track prediction, improvements in TC intensity predictions by global atmospheric models remain a challenge. One factor that impedes improvement in the intensity prediction is the lack of horizontal resolution to capture essential mechanisms of TC intensity changes. TC intensity and its variation are closely related to the inner-core structure and convective activity (e.g. Rogers et al., 2013; Wang and Wang, 2014). Recent studies using a high-resolution, limited-area atmospheric model show that the use of a horizontal resolution of a few kilometres is necessary to realistically reproduce the inner-core structure and associated convection (e.g. Braun and Tao, 2000; Gentry and Lackmann, 2010; Kanada and Wada, 2015). Fierro et al. (2009) examined the dependence of TC intensity prediction using horizontal resolutions from 30 km to 1 km and pointed out that the predicted TC intensity became increasingly realistic with resolutions between 15 km and 5 km. Therefore, the use of a high-resolution global atmospheric model with a horizontal resolution of <10 km is promising to improve TC intensity and track prediction.

### 1.3 TYMIP-G7

The primary objectives of the 'Global 7-km mesh nonhydrostatic Model Intercomparison Project for improving TYphoon forecast' (TYMIP-G7) are to understand and statistically quantify the advantages of high-resolution global atmospheric models towards the improvement of TC track and intensity forecasts. The project is conducted as a strategic program of the Earth Simulator of the Japan Agency for Marine-Earth Science and Technology (JAMSTEC). We accomplish this objective via a model intercomparison of three 7-km mesh nonhydrostatic atmospheric models (DFSM, MSSG and NICAM) and a 20-km mesh hydrostatic operational atmospheric model of the JMA (Global Spectral Model; GSM) in various cases. Because a huge amount of data is produced by each model, we developed an effective method to handle and visualize the data. Sharing the knowledge obtained in this project with research and operational communities will facilitate R2O.

In this paper, we describe the specifications of TYMIP-G7 and the set of metrics used to validate the model performances. Some preliminary results concerning the metrics are also shown. This paper comprises six sections. Section 2 describes the common experimental design, including the cases and the output dataset. Section 3 briefly overviews the scientific outcomes of each model and describes the detailed specifications. Section 4 presents the metrics, analysis method and visualization. Preliminary results concerning the advantages of high-resolution models for TC prediction and the simulated TC wind structure are given in Section 5. Section 6 is devoted to conclusions and future work.

### 2 Experimental design

We imitated JMA operational specifications to conduct 5-day numerical experiments with the models (DFSM, GSM, MSSG and NICAM). The JMA 6-hourly global objective analysis data were used for each model to derive atmospheric initial conditions. The data were provided based on the GSM grid system, a linear Gaussian grid with a horizontal resolution of 20 km and a hybrid sigma-pressure vertical coordinate. DFSM and GSM interpolated data directly onto their model grids, whereas MSSG and NICAM preliminarily interpolated the data onto common latitude/longitude grids and pressure levels and then interpolated this to their model grids. A merged satellite and in situ data global daily sea surface temperature (SST) product with a horizontal resolution of 0.25° (Kurihara et al., 2006) was used for the SST oceanic initial conditions and the sea ice concentration. Because an atmospheric model was used in the present study, SSTs for the 5-day integration were given as the boundary conditions. It was assumed that an SST anomaly from an observed daily climatology on an initial date persisted during the 5-day period. Even though no diurnal cycle of SST was input into the models, NICAM can simulate the diurnal cycle because it is coupled with a simple bulk ocean model, as described later.

The project was implemented using the Earth Simulator, a supercomputer system operated by JAMSTEC. The Earth Simulator is based on NEC SX-ACE, a distributed-memory, massively parallel vector system with a total of 5120 computational nodes. Each node has one central processing unit, which comprises four processing cores and a 64 GB main memory. The theoretical peak performance of the entire system is 1.3 peta floating-point operations per second.

## 2.1 Cases

We conducted the project for two stages: from June 2015 to September 2015 and from October 2015 to March 2016. The first stage addressed TCs from September to October in 2013, during the most active TC season since 1951. We calculated 9 TCs in 52 runs (Table 1). However, we detected some flaws in MSSG and NICAM, and we could not perform some of the numerical experiments. The second stage addressed the lifecycle of a TC, e.g. genesis, rapid intensification, recurvature, extratropical transition in addition to the Madden–Julian oscillation (MJO; Madden and Julian, 1972) and the boreal summer intraseasonal oscillation (BSISO; Wang and Rui, 1990; Wang and Xie, 1997). After we improved the detected flaws, we examined 13 TCs in 85 runs (Table 2) in addition to the numerical experiments in the first stage. We analyse the model output obtained in the second stage in this paper.

## 2.2 Dataset

Model output data for every 1 or 3 hours from each experiment (Tables 1 and 2) were stored for analyses. The components of the output are listed in Table 3. Even though each model uses its own grid system, the output data were prepared for a regular latitude/longitude (lat/lon) grid system. In TYMIP-G7, we used GrADS file formats (pairs of 4-byte IEEE 754 floating-point standard with a big endian binary file and a control file in text format) that are common in the atmospheric and oceanic research fields. The domain of the output data covers the globe, including the western North Pacific Ocean (100–180° E 0–60° N). For the MJO and BSISO cases (20 runs, see Tables 1 and 2), it also covers the Tropics (30° E–100° W 15° S–30° N). The horizontal resolution of the global dataset is 1.25°. The data for the western North Pacific Ocean and the Tropics are prepared with a horizontal resolution of ~0.07° (7 km) by DFSM, MSSG and NICAM and ~0.19° (20 km) by GSM. In the vertical direction, the data were prepared on 32 common pressure levels (in hPa: 1000, 975, 950, 925, 900, 875, 850, 825, 800, 775, 750, 700, 650, 600, 550, 500, 450, 400, 350, 300, 250, 225, 200, 175, 150, 125, 100, 70, 50, 30, 20 and 10).

## 3 Models

We used three 7-km mesh nonhydrostatic global atmospheric models in TYMIP-G7 (Fig. 1). The DFSM was developed in the MRI of the JMA. The MSSG was developed at JAMSTEC. NICAM was developed at JAMSTEC, the University of Tokyo and the RIKEN Advanced Institute for Computational Science. In addition, we used GSM with a horizontal grid spacing of ~20 km to quantify the advantages of the higher-resolution models. DFSM and GSM are spectral models and MSSG and NICAM are grid models. The following subsections detail the aforementioned models (Table 4) and the experimental design.

## 3.1 GSM and DFSM

GSM (JMA, 2013) is a hydrostatic global spectral atmospheric model using spherical harmonics. The JMA has used this model operationally to provide fundamental information for forecasts. The model was put into operation in 1988 with T63L16 resolution (200-km mesh), where 'Tx' refers to the horizontal triangular spectral truncation with a total wavenumber x using a quadratic Gaussian grid and 'Ly' refers to the number of vertical layers y. The resolution of the operational GSM increased to T106L21 (120-km mesh) in 1989, T213L30 (60-km mesh) in 1996, T213L40 in 2001, TL319L40 (60-km mesh)

in 2005, TL959L60 (20-km mesh) in 2007 and TL959L100 in 2014 (JMA, 2016), where 'TLx' refers to the horizontal triangular spectral truncation with a total wavenumber x using a linear Gaussian grid (Hortal, 2002).

The JMA has also used GSM as the principal part of an ensemble prediction system for medium-range weather forecasts. The forecast data are widely provided via the framework of The observing-system research and predictability experiment Interactive Grand Global Ensemble (TIGGE) for the research community. TIGGE data have been used for various applications, including TC track prediction (Yamaguchi et al., 2012, 2015) and the MJO (Matsueda and Endo, 2011). In addition, GSM has been used to produce atmospheric reanalysis datasets, i.e. the Japanese 25-year ReAnalysis (JRA-25; Onogi et al., 2007) and the Japanese 55-year ReAnalysis (JRA-55; Kobayashi et al., 2015). MRI global climate models have been developed based on GSM and have been used in climate research, such as global warming projections (e.g. Mizuta et al., 2006; Yukimoto et al., 2011) and stratospheric studies (e.g. Shibata et al., 1999). TC activity in future climates has been intensively studied using various model physics and horizontal resolutions (Murakami and Sugi, 2010; Murakami et al., 2012a, 2012b).

The MRI developed DFSM by changing the hydrostatic dynamical core of GSM using spherical harmonics to a nonhydrostatic dynamical core using a double Fourier series (Yoshimura, 2012). DFSM uses the same basis functions of the double Fourier series as Cheong (2000). In DFSM, a fast Fourier transform is used instead of a Legendre transform in the meridional direction. Because the computational cost of the fast Fourier transform is much smaller than that of the Legendre transform, especially at high resolution, DFSM is applicable to finer resolution simulations. DFSM gives nearly the same results as GSM using the Legendre transform; a comparison of 2-day forecasts using the 60-km resolution model was shown by Yoshimura and Matsumura (2005).

In GSM and DFSM, a two-time-level, semi-implicit, semi-Lagrangian scheme (e.g. Hortal, 2002) is used to facilitate long time steps for computational efficiency. The vertically conservative semi-Lagrangian scheme is used in the advection calculation (Yoshimura and Matsumura, 2003; Yoshimura and Matsumura, 2005; Yukimoto et al., 2011), and a correction method similar to that described by Priestley (1993) and Gravel and Staniforth (1994) is used for global conservation in the material transport. To save computational costs, we used a reduced grid (Miyamoto, 2006) in which the number of zonal grid points is decreased, especially at high latitudes (Fig. 1).

Because the DFSM resolution is ~7 km (ML2559L100; 'MLx' refers to a horizontal truncation with zonal wavenumber x using a linear equally-spaced latitude grid), the model applies the nonhydrostatic option, which essentially uses the same nonhydrostatic equations as used in the ALADIN-NH nonhydrostatic limited-area spectral model (Bubnová et al., 1995; Bénard et al., 2010) and the nonhydrostatic version of the Integrated Forecast System global model of ECMWF (Wedi and Smolarkiewicz, 2009). However, there are some differences in the method of integration. DFSM uses a non-constant coefficient semi-implicit scheme. The preconditioned Generalized Conjugate Residual method, a fast-converging iteration method, is used to solve the simultaneous linear equations associated with the semi-implicit scheme (Yoshimura, 2012). Recalculation is necessary only for the non-constant linear terms during the iteration. It is found that only a single iteration is sufficient for convergence.

Physical packages included in GSM and DFSM are the same as those in the March 2014 version of the operational global atmospheric model of the JMA. A prognostic cumulus parameterization scheme (Randall and Pan, 1993) and other schemes in GSM are used in DFSM without any changes. The physical process is described in detail in the JMA (2013).

## 3.2 MSSG

MSSG is an atmosphere–ocean coupled nonhydrostatic model aimed at a seamless simulation from global to local scales (Takahashi et al., 2006, 2013). The MSSG comprises atmospheric (MSSG-A) and oceanic (MSSG-O) components. MSSG uses a conventional lat/lon grid system for regional simulations and the Yin–Yang grid system (Kageyama and Sato, 2004; Baba et al., 2010), which comprises two overlapping lat/lon grids to avoid the polar singularity problem, for global

simulations. MSSG has been used in a wide range of applications. A cloud-system-resolving global ocean–atmosphere coupled MSSG successfully simulated an observed MJO propagation (Sasaki et al., 2016). A global atmosphere–ocean coupled experiment with 11-km horizontal resolution with a nested region with 2.7-km horizontal resolution simulated sea surface cooling caused by a TC along its track (Takahashi et al., 2013). High-resolution regional atmospheric simulations

have been conducted to investigate the influence of the choice of cloud microphysics scheme and in-cloud turbulence on cloud development (Onishi et al., 2011, 2012). MSSG-O with a 2-km horizontal resolution has been used to investigate the dispersion of radionuclides released from the Fukushima Daiichi nuclear power plant (Choi et al., 2013) and the effect of wind on long-term summer water temperature trends in Tokyo Bay, Japan, with 200-m horizontal resolution (Lu et al., 2015). MSSG-A with a 5-m spatial resolution has been used in building-resolving urban atmosphere simulations to examine the

heat environments of streets (Takahashi et al., 2013).

In this study, MSSG-A is primarily used. Its dynamical core is based on the nonhydrostatic equations, and it predicts the three wind components, air density and pressure. Each horizontal computational domain covers $4056 \times 1352$ grids in the Yin–Yang lat/lon grid system. The average horizontal grid spacing is 7 km. The vertical level comprises 55 vertical layers with a top height of 40 km and the lowermost vertical layer at 75 m. The third-order Runge–Kutta scheme is

15 used for time integration. The fast terms related to acoustic and gravity waves are calculated separately with a shorter time step (Wicker and Skamarock, 2002). A fifth-order upwind scheme (Wicker and Skamarock, 2002) was chosen for the momentum advection and a second-order weighted average flux scheme with the Superbee flux limiter (Toro, 1989) for the scalar advection. For turbulent diffusion, the Mellor–Yamada–Nakanishi–Niino level 2.5 scheme (Nakanishi and Niino, 2009) was used. The MSSG-Bulk model (Onishi and Takahashi, 2012), a six-category bulk cloud microphysics model, is

20 used for explicit cloud physics. Model Simulation radiation TRaNsfer code version 10 (MstrnX; Sekiguchi and Nakajima, 2008) is used to calculate longwave and shortwave radiation transfer.

During the first stage of the project, extraordinary increases in precipitable water appeared in the 5-day integrations when the conventional bulk surface flux model of Zhang and Anthes (1982) was used for both land and ocean surfaces. This issue was solved by the use of the COARE 3.0 model (Fairall et al., 1996, 2003) for ocean surface fluxes with Zhang and

25 Anthes (1982) being used only for land surface fluxes. This combination was used for all simulations in the second stage, and we plan to rerun all the simulations in the first stage.

### 3.3 NICAM

NICAM (Satoh et al., 2008, 2014) was developed as a climate model and can explicitly resolve clouds without any

convective parameterization, which is known to be the most ambiguous component in conventional climate models (Randall et al., 2003). From the first appearance of realistic cloud-resolving simulations using a 3.5-km-mesh horizontal resolution by Miura et al. (2007a), NICAM has primarily been used to study tropical meteorological systems, such as the MJO (Miura et al., 2007b, Nasuno, 2013; Miyakawa et al., 2014), TC genesis from the MJO in boreal winter (Fudeyasu et al., 2008, 2010a, 2010b), TC genesis from the BSISO in the western North Pacific (Oouchi et al., 2009; Nakano et al., 2015; Nasuno et al.

2016) and BSISO in the northern Indian Ocean (Taniguchi et al., 2010; Yanase et al., 2010). NICAM has also been used for quasi-real-time forecast systems during field observation campaigns to support field observations (Nasuno, 2013). Recent progress with high-performance computing infrastructures, such as the K-computer, a 10-petaflop supercomputer in Japan, facilitates 870-m mesh global simulations (Miyamoto et al., 2013, 2015; Kajikawa et al., 2016). This is the highest resolution to date (10 July 2016). Climate simulations (30-year) using a 14-km mesh model (Kodama et al., 2015) and large-member

(10240 members) ensemble data assimilations based on an ensemble Kalman filter (Miyoshi et al., 2015) have also been executed.

NICAM uses an icosahedral grid system that covers the globe with a nearly uniform grid size, avoiding the polar singularity problem. Increased horizontal resolution is attained by recursively dividing horizontal grids in half. Therefore, the possible horizontal resolution is discrete and represented by the 'g-level', which indicates the number of divisions of a horizontal grid. In this project, the 2014 version of NICAM (called NICAM.14.3) was used with a horizontal resolution of g-level 10, corresponding to a 7-km mesh. The vertical level comprises 38 vertical layers to a top height of 36.7 km with the lowest layer at 80 m. NICAM uses a fully compressible nonhydrostatic equation system for the dynamics of the atmosphere. The model uses an icosahedral grid system in the horizontal direction with the Arakawa A-grid and terrain-following coordinate with the Lorenz grid in the vertical direction. The equations are discretized using the flux form of the finite volume method. The numerical scheme guarantees conservation of total mass and energy. The second-order Runge–Kutta scheme is primarily used for time integration, whereas the third-order Runge–Kutta scheme is used in some cases to avoid computational instability. NICAM uses the split-explicit scheme together with the horizontal explicit and vertical implicit scheme to avoid the restriction of the Courant–Friedrichs–Lewy condition for acoustic waves. The NICAM Single-moment Water 6 cloud microphysics scheme (Tomita, 2008) is used for cloud microphysics without any convective parameterization. Planetary boundary layer processes are calculated using the Mellor–Yamada–Nakanishi–Niino level 2 scheme (Nakanishi and Niino, 2004) implemented and examined by Noda et al. (2010). Longwave and shortwave radiation transfer is calculated using MstrnX (Sekiguchi and Nakajima, 2008). Land surface processes are computed by the Minimal Advanced Treatments of Surface Interaction and Runoff (MATSIRO; Takata et al., 2003). NICAM is coupled with a simple slab ocean model. This model calculates SST based on the local heat balance between the ocean slab and the atmosphere, and the other ocean dynamics, such as vertical mixing and advection, are not considered. The slab has a specific heat capacity determined by its thickness (15 m). The calculated SST is nudged with a persistent SST anomaly with an e-folding time of 7 days. The surface flux is calculated by the Louis (1979) scheme with sea surface roughness length parameterization by Moon et al. (2007).

During the first stage of this project, there were frequent problems of divisions by zero in MATSIRO that had not been experienced in simulations with coarser horizontal resolutions. This issue was fixed before simulations in the second stage, and abnormal cases in the first stage had to be rerun. The fix had a slight impact on the prediction results. During the second stage, however, two cases were still unable to be completed due to numerical instability (Table 2).

## 4 Metrics, analysis methods and visualization

### 4.1 Metrics

Here, we define the following metrics to evaluate the TC forecast performance.

(1) Computational resources for a 5-day forecast on the Earth simulator (node-hours)

(2) TC track (position) error every 6 hours of forecast time (km)

(3) TC intensity (central pressure) error every 6 hours of forecast time (hPa)

(4) Averaged radius of surface 50-knot (25 m s$^{-1}$) wind (AR50) error (km)

(5) Averaged radius of surface 30-knot (15 m s$^{-1}$) wind (AR30) error (km).

It is important for the operational model that the calculation is completed in less time with smaller computational resources so that we applied metric (1). The metrics (2)–(5) measure the accuracy of the track, intensity and surface wind structure prediction based on the RSMC Tokyo best-track data.

### 4.2 TC tracking

We extract TC tracks from the model experiments using the hourly mean sea level pressure (SLP) data with a horizontal resolution of ~7 km for DFSM, MSSG and NICAM and 20 km for GSM. A TC centre is defined as a minimum

SLP point from the predicted mean SLP field smoothed 100 times by a 1-2-1 filter for each longitude and latitude. The initial TC centre is defined within a radius of 1° from a centre position based on the RSMC Tokyo best-track data. The next centre position is defined as the minimum SLP point from the smoothed SLP field within a radius of 1° from the previous centre position. The tracking terminates when the minimum SLP points reach a proximity of 1° from the lateral boundary in the domain of the output data. The tracking algorithm works well for nearly all cases; however, misdetection occurred for some very weak TCs. These cases were excluded from the validation.

### 4.3 AR50 and AR30

The RSMC Tokyo best-track data contains longest and shortest radii of 50-knot and 30-knot wind speeds and their direction. AR50 and AR30 are defined as the average of the longest and shortest radii of the 50-knot and 30-knot wind speeds, respectively. The directions of the longest and shortest radii are defined by eight directions (N, NE, E, SE, S, SW, W and NW) in the best-track data. Therefore, we calculated the radii of the 50-knot and 30-knot wind in the model in each of the eight directions first and then determined the direction of the longest and shortest radii. Then, the radii in those two directions were averaged to obtain AR50 and AR30.

### 4.4 Multi-model ensemble mean

The multi-model ensemble mean (MME) is applied to the three 7-km mesh models (DFSM, MSSG and NICAM). MME is a simple ensemble average derived from a combination of individual models, which reduces the average forecast error relative to the best individual predictions by the individual models. MME also provides additional information about the forecast uncertainty, enhancing forecast confidence (Goerss, 2000; Yamaguchi et al., 2012).

### 4.5 Visualization

We developed a Web application that allows the simultaneous visualization of multi-model results. Figure 2 shows a screen capture of this application, which portrays digital globes using Cesium.js (Analytical Graphics, Inc., 2015), a WebGL-based virtual globe and map engine. Visualization results of each model are overlaid on them. We used Volume Data Visualizer for Google Earth (VDVGE; Kawahara, 2012; Kawahara, 2015) to depict visualization results for the overlay. VDVGE is originally a visualization software that exports visualization results in the KML format, a data format suitable for Google Earth. An option to export in the CZML format, suitable for Cesium.js, has recently been implemented in VDVGE. The present Web application enables us to view the animation display for time-series visualization results of each model while synchronously changing the three-dimensional viewpoint. An option to display each model result selectively is also available. This application enables the four models to be easily compared.

## 5 Results

### 5.1 Computational resources

Computational performance is an important metric for an operational numerical weather forecast model. DFSM, MSSG, and NICAM models consumed computational resources equivalent to 682, 2330, and 1155 node-hours, respectively for a case on 12 September 2013, 00:00:00 UTC. These quantities did not vary greatly between cases because the computational nodes were occupied in each calculation and the disk I/O was executed from/to the work disk mounted on each computational node. Note that the computational resources required for each model are highly dependent on the model specifications (e.g. the physics scheme, advection scheme, number of vertical layers, vertical resolution and time step) and the degree of optimization for the Earth Simulator.

## 5.2 Track predictions

To quantify the advantage of using finer resolutions for TC track prediction, we examined the time series of TC track prediction errors with reference to the RSMC Tokyo best track for the second stage (Figure 3). TC track predictions by DFSM, MSSG and NICAM performed better than GSM. However, the reduction in the track errors depended on the TC case. That is, the use of finer resolution alone does not always improve TC track prediction. This suggests that improvements in the initial conditions and that of the physical processes in each model are also required to improve track prediction.

We also validated MME using track predictions of the three models with reference to the RSMC Tokyo best-track data. MME track prediction gave the smallest track errors for forecast time (FT) of 24–120 hours. The reduction rate of the MME position error from that of GSM was ~26% at FT = 120 hours relative to that of GSM. The position error of MME at that FT corresponds to that of GSM at FT = 96 hours. Even though MME had promising results with regard to improving TC track prediction, future work is required to achieve more robust results and to answer scientific and practical questions, such as 'in which cases is MME effective and why?'

## 5.3 Intensity predictions

Figure 4 shows time series of the average central pressure and the standard deviation in each model relative to the RSMC Tokyo best-track data for the second stage. Because the global objective analysis data, which was used as initial conditions of the numerical experiments, tend to reproduce TC central pressure shallower than those in RSMC Tokyo best-track data, cases with an initial bias <20 hPa are validated. The central pressures in MSSG and NICAM showed relatively small biases compared to the error in GSM. These results indicate that these 7-km mesh models help decrease systematic positive errors for the central pressure. However, the central pressure in DFSM showed over-intensification and the magnitude of the bias after FT = 54 hours became larger than that in GSM. Because both DFSM and GSM had the same specifications except for the horizontal resolution, this result suggests that the improvement on of physics schemes suitable for such high-resolution models are needed for accurate forecasts of the central pressure. GSM showed a gradual growth of positive bias in the central pressure until FT = 84 hours, including the initial 24 hours, when the 7-km mesh models showed a continuous reduction in the errors. After this early reduction, the errors of the 7-km mesh models began to grow in model-specific ways. MSSG showed a gradual growth of positive bias in the central pressure until FT = 84 hours and then the errors become saturated. NICAM retained nearly no bias for the central pressure until FT = 84 hours and then showed a slight growth in the negative bias for the central pressure until FT = 120 hours. DFSM had a gradual growth of negative bias for the central pressure until FT = 120 hours. MME showed a negative bias for the central pressure after FT = 24 hours.

## 5.4 Predictions of the TC wind structure

Accurate predictions of AR50 and AR30 lead to accurate estimations of the area affected by TCs. Figure 5 shows the validation result of AR50 based on the RSMC Tokyo best-track data. All models had negative bias of 80–90 km even at the initial time. This negative bias is partially attributed to the shallower estimation of the central pressure by ~5 hPa (Figure 4) associated with the biases in the global objective analysis data, which was used as initial conditions of the numerical experiments. The difference in the interpolation methods to prepare the initial data for each model might also affect the bias. The negative biases of all 7-km models decrease in the early stage. The negative bias of DFSM monotonically decreases until FT = 78 hours and then saturates at ~25 km at FT = 78–120 hours. The bias of MSSG decreases more rapidly until FT = 48 hours and becomes positive until FT = 84 hours and then returns to a negative bias of ~20 km. The bias of NICAM continuously decreases until FT = 66 hours and then becomes positive. At FT = 120 hours, NICAM shows a positive bias of 40 km, which was a smaller magnitude than that of the initial bias. Conversely, GSM shows little improvement in the negative bias so that its negative bias remains at ~60 km at FT = 120 hours. These results show that high-resolution models

can significantly reduce the error of AR50. In addition, MME has a promising result in improving the AR50 prediction: MME showed a bias of nearly zero for FT = 60–120 hours.

Figure 6 shows the validation results of AR30. All models show a negative bias of more than 200 km at FT = 0 hours. The negative biases of all 7-km models tended to decrease in the early stage as FT proceeded. The negative bias of DFSM decreases to 180 km by FT = 36 hours and then relatively slowly decreases to 150 km by FT = 120 hours. The negative bias of MSSG temporarily increases in the first 6 hours, and then decreases. The bias of NICAM continuously decreases up to FT = 120 hours, resulting in a negative bias as small as 35 km at FT = 120 hours. GSM had little improvement in AR30 up to FT = 96 hours and shows a negative bias of ~170 km at FT = 120 hours. These results show that high-resolution models can also reduce the error in AR30. However, all the models still had relatively large negative biases compared to the error in AR50. Towards a better prediction of TC wind structure, further improvements in the quality of the objective analysis and the models themselves are needed. The bias of MME also decreases up to FT = 120 hours; however, its magnitude is larger than that of NICAM.

An accurate prediction of the three-dimensional TC structure can lead to accurate predictions of the intensity, AR30 and AR50. Because there is no high-resolution TC observation that is suitable for the validation of the simulated TC structure, here we made an intercomparison of the TC wind structures simulated by the 7-km models and 20-km mesh GSM. Figure 7 shows a composite of the radius-height section of the azimuthal mean radial and tangential wind speeds for TCs at the time of the RSMC Tokyo best-track central pressure between 920–940 hPa, corresponding, in the lifecycle, to the mature stage of a TC. A total of 347 snapshots were used for the composite analysis. If the models can simulate the TC structure perfectly, the result should be the same for all models. While all 7-km mesh models reproduced typical axisymmetric mean inner-core structures, such as primary and secondary circulations, the simulated TC structures differed significantly between the 7-km models. The TCs calculated by DFSM had the highest maximum tangential wind speed and the smallest radius of maximum wind (RMW) of the models. In addition, its primary circulation was the deepest, reaching up to 100 hPa in the vertical direction and the narrowest in the horizontal direction. The depth of the inflow and outflow layers in DFSM was relatively thin and had the strongest radial velocity. The TCs in NICAM and MSSG showed relatively similar structures to each other; however, MSSG had thicker inflow and outflow layers. Differences in the heating and inertial stability in the inner-core lead to differences in the primary and secondary circulation (Shapiro and Willoughby 1982). Understanding the cause of the differences in the simulated structures in the models will lead to improvements in all the models.

## 6 Conclusions and future work

TYMIP-G7 was implemented in two stages from June 2015 through March 2016. The aim of the project was to statistically quantify and understand the advantages of high-resolution, global atmospheric models to improve 5-day TC track, intensity and wind radii forecasts. We performed numerical experiments for multiple TC cases in 137 runs using three 7-km mesh global nonhydrostatic atmospheric models: DFSM, MSSG and NICAM. We also included a 20-km mesh global hydrostatic atmospheric model, GSM, on the Earth Simulator of JAMSTEC. We statistically evaluated errors in the TC track, intensity and wind radii predictions with the following primary results.

(C1) The 7-km models statistically improve both the TC intensity and track predictions, whereas the improvement in the individual TC tracks depends on the case.

(C2) The MME is a promising approach to further enhance the TC track and AR50 predictions.

(C3) The predicted TC structure differs greatly between the three models even though they have the same horizontal resolution.

To follow up the above results to further improve TC prediction, we must answer the following questions.

(Q1) Why are the TC predictions improved by high-resolution models?

(Q2) What causes the differences in the simulated TC structure in the three 7-km mesh atmospheric global models, such as the radius of the maximum winds, the eyewall slope, the inflow and outflow layers and the rainbands?

To answer (Q1), an intercomparison of forecasts by the 20-km mesh GSM and the 7-km mesh models (DFSM, MSSG and NICAM) is the first step. Concerning (Q2), the predicted TC structure depends on the physics schemes, such as cloud microphysics, planetary boundary layer and surface flux, as well as the dynamical core of the model. To understand the impacts of the model physics schemes, sensitivity experiments altering the schemes and/or tuning parameters will be required.

In addition, the following topics are suggested for future work:

(F1) Extended-range forecasts, contributing to TC genesis and MJO/BSISO forecasts;

(F2) Atmosphere–ocean coupled experiments to examine impacts on TC intensity and track and MJO/BSISO;

(F3) Further high-resolution experiments to study impacts of better inner-core representation on TC intensities and tracks; and

(F4) Data assimilation to contribute for validating the models and understanding the TC processes and model initializations. These topics are addressed below.

An advantage of global models for TC prediction over limited-area models is the coverage of multi-scale atmospheric phenomena from a mesoscale vortex to synoptic environments. Because TC genesis strongly depends on synoptic environments modulated by the MJO/BSISO, global models should be used for its forecasting. Indeed, Nakano et al. (2015) and Xiang et al. (2015) showed that TC genesis is predictable up to two weeks in advance; this great skill in TC genesis forecasting was attributed to its strong ability to forecast BSISO/MJO. We are conducting extended-range (longer than two weeks) forecast experiments using the four models in several cases and will investigate the advantage of high-resolution modes.

In the present project, atmosphere models were used, except for NICAM, which is coupled with a simple slab ocean model. However, studies have shown that fully coupled atmosphere–ocean processes are essential for especially slow-moving, intense TCs (Yablonsky and Genis, 2009). These processes affect the TC structure and therefore the track and intensity. In addition, aa fully coupled atmosphere–ocean model is better for MJO/BSISO forecasts. MSSG is already capable of coupling MSSG-A with MSSG-O (Sasaki et al., 2016; Takahashi et al., 2013). In addition, NICAM has been coupled with the Center for Climate System Research Ocean COmponent Model (COCO; Hasumi, 2006). Therefore, we will use these coupled global models to examine the impacts of global atmosphere–ocean processes on TC forecasts.

To improve the high-resolution models, the validation of simulated phenomena using observations is essential. An understanding of the essential processes and the modelling therefore requires high-resolution spatiotemporal observations. Recent advances in satellite observations furnish quantitatively and qualitatively rich observational data. However, the spatiotemporal resolution is still insufficient for the validation of TC structures simulated by high-resolution models. Aggressively developing data assimilation techniques using satellite observations (e.g. Zhang et al., 2016, Okamoto et al., 2016) is a promising means of obtaining high-resolution, spatiotemporal, three-dimensional TC structures, including at the cloud convection scale (~O(1 km)). In addition, applying such cloud-resolving analyses to deriving the initial conditions of high-resolution models may improve TC prediction.

**Data availability**

The initial and boundary data for the models and model outputs are available under a collaborative framework between MRI, JAMSTEC and related institutes or universities.

**Acknowledgements**

This project was conducted as 'The Earth Simulator Strategic Project with Special Support' of JAMSTEC. All numerical experiments were run on the Earth Simulator (NEC SX-ACE). This study was partly supported by HPCI Strategic Programs for Innovative Research (SPIRE) Field 3, the FLAGSHIP 2020 project of the Ministry of Education, Culture, Sports, Science and Technology (MEXT) and KAKENHI 26282111, 26400475 and 15K05292 of the Japan Society for the Promotion of Science (JSPS). The authors thank Ms Mikiko Ikeda, Mr Yuichi Saitoh and Mr Hiromitsu Fuchigami for supporting the experiments on the Earth Simulator. The authors also acknowledge Mr Hideaki Kawai and Mr Eiki Shindo for the fruitful discussions. The schematic diagram of the NICAM grid was provided by Professor Masaki Satoh.

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

**Table 1. List of initial times for stage 1 of TYMIP-G7.**

| | Initial time | Typhoon case (*Italic*: weaker than Tropical Storm, ***Bold italic***: ***extratropical cyclone***) | DFSM | GSM | MSSG | NICAM |
|---|---|---|---|---|---|---|
| 1 | 12 September 2013, 00:00:00 UTC | *Man-yi* | ○ | ○ | ○ | ○ |
| 2 | 12 September 2013, 06:00:00 UTC | *Man-yi* | ○ | ○ | ○ | ○ |
| 3 | 12 September 2013, 12:00:00 UTC | *Man-yi* | ○ | ○ | ○ | ○ |
| 4 | 12 September 2013, 18:00:00 UTC | *Man-yi* | ○ | ○ | ○ | ○ |
| 5 | 13 September 2013, 00:00:00 UTC | Man-yi | ○ | ○ | ○ | ○ |
| 6 | 30 September 2013, 00:00:00 UTC | Wutip, Sepat, *Fitow* | ○ | ○ | ○ | ○ |
| 7 | 30 September 2013, 06:00:00 UTC | Wutip, Sepat, *Fitow* | ○ | ○ | ○ | ○(*1) |
| 8 | 30 September 2013, 12:00:00 UTC | Wutip, Sepat, *Fitow* | ○ | ○ | ○ | ○ |
| 9 | 30 September 2013, 18:00:00 UTC | Wutip, Sepat, *Fitow* | ○ | ○ | ○ | ○ |
| 10 | 1 October 2013, 00:00:00 UTC | ***Wutip***, Sepat, Fitow, *Danas* | ○ | ○ | ○ | ○ |
| 11 | 1 October 2013, 06:00:00 UTC | Sepat, Fitow, *Danas* | ○ | ○ | ○ | ○(*1) |
| 12 | 1 October 2013, 12:00:00 UTC | Sepat, Fitow, *Danas* | ○ | ○ | ○ | ○(*1) |
| 13 | 1 October 2013, 18:00:00 UTC | Sepat, Fitow, *Danas* | ○ | ○ | ○ | ○(*1) |
| 14 | 2 October 2013, 00:00:00 UTC | Sepat, Fitow, *Danas* | ○ | ○ | ○ | ○(*1) |
| 15 | 2 October 2013, 06:00:00 UTC | Sepat, Fitow, *Danas* | ○ | ○ | ○ | ○ |
| 16 | 2 October 2013, 12:00:00 UTC | Sepat, Fitow, *Danas* | ○ | ○ | ○ | ○ |
| 17 | 2 October 2013, 18:00:00 UTC | ***Sepat***, Fitow, *Danas* | ○ | ○ | ○ | ○(*1) |
| 18 | 3 October 2013, 00:00:00 UTC | ***Sepat***, Fitow, *Danas* | ○ | ○ | ○ | ○ |
| 19 | 3 October 2013, 06:00:00 UTC | ***Sepat***, Fitow, *Danas* | ○ | ○ | ○ | ○ |
| 20 | 3 October 2013, 12:00:00 UTC | ***Sepat***, Fitow, *Danas* | ○ | ○ | ○ | ○ |
| 21 | 3 October 2013, 18:00:00 UTC | ***Sepat***, Fitow, *Danas* | ○ | ○ | ○ | ○(*1) |
| 22 | 4 October 2013, 00:00:00 UTC | Fitow, *Danas* | ○ | ○ | ○ | ○(*1) |
| 23 | 9 October 2013, 00:00:00 UTC | ***Danas***,*Nari*,*Wipha* | ○ | ○ | ○ | ○ |
| 24 | 9 October 2013, 06:00:00 UTC | ***Danas***,*Nari*,*Wipha* | ○ | ○ | ○ | ○ |
| 25 | 9 October 2013, 12:00:00 UTC | Nari,*Wipha* | ○ | ○ | ○ | ○ |
| 26 | 9 October 2013, 18:00:00 UTC | Nari,*Wipha* | ○ | ○ | ○ | ○ |
| 27 | 10 October 2013, 00:00:00 UTC | Nari,*Wipha* | ○ | ○ | ○ | ○ |
| 28 | 10 October 2013, 06:00:00 UTC | Nari,*Wipha* | ○ | ○ | ○ | ○ |
| 29 | 10 October 2013, 12:00:00 UTC | Nari,Wipha | ○ | ○ | ○ | ○ |
| 30 | 10 October 2013, 18:00:00 UTC | Nari,Wipha | ○ | ○ | ○ | ○ |
| 31 | 11 October 2013, 00:00:00 UTC | Nari,Wipha | ○ | ○ | ○ | ○ |
| 32 | 11 October 2013, 06:00:00 UTC | Nari,Wipha | ○ | ○ | ○ | ○ |
| 33 | 11 October 2013, 12:00:00 UTC | Nari,Wipha | ○ | ○ | ○ | ○ |
| 34 | 11 October 2013, 18:00:00 UTC | Nari,Wipha | ○ | ○ | ○ | ○ |
| 35 | 12 October 2013, 00:00:00 UTC | Nari,Wipha | ○ | ○ | ○ | ○ |
| 36 | 12 October 2013, 06:00:00 UTC | Nari,Wipha | ○ | ○ | ○ | ○ |
| 37 | 12 October 2013, 12:00:00 UTC | Nari,Wipha | ○ | ○ | ○ | ○ |
| 38 | 17 October 2013, 12:00:00 UTC | ***Wipha***, Francisco | ○ | ○ | ○ | ○ |
| 39 | 17 October 2013, 18:00:00 UTC | ***Wipha***, Francisco | ○ | ○ | ○ | ○ |
| 40 | 18 October 2013, 00:00:00 UTC | ***Wipha***, Francisco | ○ | ○ | ○ | ○ |
| 41 | 18 October 2013, 06:00:00 UTC | ***Wipha***, Francisco | ○ | ○ | ○ | ○ |
| 42 | 18 October 2013, 12:00:00 UTC | ***Wipha***, Francisco | ○ | ○ | ○ | ○(*1) |
| 43 | 18 October 2013, 18:00:00 UTC | Francisco | ○ | ○ | ○ | ○ |

| 44 | 19 October 2013, 00:00:00 UTC | Francisco, *Lekima* | ○ | ○ | ○ | ○(*1) |
| 45 | 19 October 2013, 06:00:00 UTC | Francisco, *Lekima* | ○ | ○ | ○ | ○ |
| 46 | 19 October 2013, 12:00:00 UTC | Francisco, *Lekima* | ○ | ○ | ○ | ○ |
| 47 | 19 October 2013, 18:00:00 UTC | Francisco, *Lekima* | ○ | ○ | ○ | ○ |
| 48 | 20 October 2013, 00:00:00 UTC | Francisco, *Lekima* | ○ | ○ | ○ | ○ |
| 49 | 20 October 2013, 06:00:00 UTC | Francisco, *Lekima* | ○ | ○ | ○ | ○ |
| 50 | 20 October 2013, 12:00:00 UTC | Francisco, *Lekima* | ○ | ○ | ○ | ○(*1) |
| 51 | 20 October 2013, 18:00:00 UTC | Francisco, *Lekima* | ○ | ○ | ○ | ○ |
| 52 | 21 October 2013, 00:00:00 UTC | Francisco, *Lekima* | ○ | ○ | ○ | ○ |

(*1): rerun with the fixed version of MATSIRO (Sect. 2.2.3).

**Table 2. List of initial times for stage 2 of TYMIP-G7.**

| | Initial time | Typhoon case (*Italic*: weaker than Tropical Storm, ***Bold italic***: ***extratropical cyclone***) and MJO/BSISO case | DFSM | GSM | MSSG | NICAM (*2) |
|---|---|---|---|---|---|---|
| 1 | 6 June 2013, 12:00:00 UTC | *Yagi* | ○ | ○ | ○ | ○ |
| 2 | 7 June 2013, 00:00:00 UTC | *Yagi* | ○ | ○ | ○ | ○ |
| 3 | 7 June 2013, 12:00:00 UTC | *Yagi* | ○ | ○ | ○ | ○ |
| 4 | 8 June 2013, 00:00:00 UTC | *Yagi* | ○ | ○ | ○ | ○ |
| 5 | 8 June 2013, 12:00:00 UTC | Yagi | ○ | ○ | ○ | ○ |
| 6 | 9 June 2013, 00:00:00 UTC | Yagi | ○ | ○ | ○ | ○ |
| 7 | 9 June 2013, 12:00:00 UTC | Yagi | ○ | ○ | ○ | ○ |
| 8 | 10 June 2013, 00:00:00 UTC | Yagi | ○ | ○ | ○ | ○ |
| 9 | 10 June 2013, 12:00:00 UTC | Yagi | ○ | ○ | ○ | ○ |
| 10 | 11 June 2013, 00:00:00 UTC | Yagi | ○ | ○ | ○ | ○ |
| 11 | 3 November 2013, 00:00:00 UTC | Krosa | ○ | ○ | ○ | ○ |
| 12 | 3 November 2013, 12:00:00 UTC | Krosa, *Haiyan* | ○ | ○ | ○ | ○ |
| 13 | 4 November 2013, 00:00:00 UTC | Krosa, Haiyan | ○ | ○ | ○ | ○ |
| 14 | 4 November 2013, 12:00:00 UTC | ***Krosa***, Haiyan | ○ | ○ | ○ | ○ |
| 15 | 5 November 2013, 00:00:00 UTC | Haiyan | ○ | ○ | ○ | ○ |
| 16 | 5 November 2013, 12:00:00 UTC | Haiyan | ○ | ○ | ○ | ○ |
| 17 | 6 November 2013, 00:00:00 UTC | Haiyan | ○ | ○ | ○ | ○ |
| 18 | 6 November 2013, 12:00:00 UTC | Haiyan | ○ | ○ | ○ | ○ |
| 19 | 7 November 2013, 00:00:00 UTC | Haiyan | ○ | ○ | ○ | ○ |
| 20 | 27 July 2014, 12:00:00 UTC | *Halong* | ○ | ○ | ○ | ○ |
| 21 | 28 July 2014, 00:00:00 UTC | *Halong* | ○ | ○ | ○ | ○ |
| 22 | 28 July 2014, 12:00:00 UTC | *Halong*, *Nakri* | ○ | ○ | ○ | ○ |
| 23 | 29 July 2014, 00:00:00 UTC | Halong, *Nakri* | ○ | ○ | ○ | ○ |
| 24 | 29 July 2014, 12:00:00 UTC | Halong, Nakri | ○ | ○ | ○ | ○ |
| 25 | 30 July 2014, 00:00:00 UTC | Halong, Nakri | ○ | ○ | ○ | ○ |
| 26 | 30 July 2014, 12:00:00 UTC | Halong, Nakri | ○ | ○ | ○ | ○ |
| 27 | 31 July 2014, 00:00:00 UTC | Halong, Nakri | ○ | ○ | ○ | ○ |
| 28 | 31 July 2014, 12:00:00 UTC | Halong, Nakri | ○ | ○ | ○ | ○ |
| 29 | 1 August 2014, 00:00:00 UTC | Halong, Nakri | ○ | ○ | ○ | ○ |
| 30 | 1 August 2014, 12:00:00 UTC | Halong, Nakri | ○ | ○ | ○ | ○ |
| 31 | 2 August 2014, 00:00:00 UTC | Halong, Nakri | ○ | ○ | ○ | ○ |
| 32 | 2 August 2014, 12:00:00 UTC | Halong, Nakri | ○ | ○ | ○ | ○ |
| 33 | 3 August 2014, 00:00:00 UTC | Halong, Nakri | ○ | ○ | ○ | ○ |
| 34 | 3 August 2014, 12:00:00 UTC | Halong, *Nakri* | ○ | ○ | ○ | ○ |
| 35 | 4 August 2014, 00:00:00 UTC | Halong, *Nakri* | ○ | ○ | ○ | ○ |
| 36 | 4 August 2014, 12:00:00 UTC | Halong | ○ | ○ | ○ | × |
| 37 | 5 August 2014, 00:00:00 UTC | Halong | ○ | ○ | ○ | × |
| 38 | 5 August 2014, 12:00:00 UTC | Halong | ○ | ○ | ○ | ○ |
| 39 | 6 August 2014, 00:00:00 UTC | Halong | ○ | ○ | ○ | ○ |
| 40 | 6 August 2014, 12:00:00 UTC | Halong | ○ | ○ | ○ | ○ |
| 41 | 7 March 2015, 00:00:00 UTC | MJO | ○ | ○ | ○ | ○ |
| 42 | 7 March 2015, 12:00:00 UTC | MJO | ○ | ○ | ○ | ○ |
| 43 | 8 March 2015, 00:00:00 UTC | MJO | ○ | ○ | ○ | ○ |
| 44 | 8 March 2015, 12:00:00 UTC | MJO | ○ | ○ | ○ | ○ |
| 45 | 9 March 2015, 00:00:00 UTC | MJO | ○ | ○ | ○ | ○ |
| 46 | 9 March 2015, 12:00:00 UTC | MJO, Pam | ○ | ○ | ○ | ○ |
| 47 | 10 March 2015, 00:00:00 UTC | MJO, Pam | ○ | ○ | ○ | ○ |
| 48 | 10 March 2015, 12:00:00 UTC | MJO, *Bavi*, Pam | ○ | ○ | ○ | ○ |
| 49 | 11 March 2015, 00:00:00 UTC | MJO, *Bavi*, Pam | ○ | ○ | ○ | ○ |
| 50 | 11 March 2015, 12:00:00 UTC | MJO, Bavi, Pam | ○ | ○ | ○ | ○ |
| 51 | 27 June 2015, 00:00:00 UTC | BSISO | ○ | ○ | ○ | ○ |
| 52 | 27 June 2015, 12:00:00 UTC | BSISO | ○ | GSM | MSSG | ○ |
| 53 | 28 June 2015, 00:00:00 UTC | BSISO | ○ | ○ | ○ | ○ |
| 54 | 28 June 2015, 12:00:00 UTC | BSISO | ○ | ○ | ○ | ○ |

| | | | | | | |
|---|---|---|---|---|---|---|
| 55 | 29 June 2015, 00:00:00 UTC | BSISO | ○ | ○ | ○ | ○ |
| 56 | 29 June 2015, 12:00:00 UTC | BSISO,*Chan-hom* | ○ | ○ | ○ | ○ |
| 57 | 30 June 2015, 00:00:00 UTC | BSISO,*Chan-hom* | ○ | ○ | ○ | ○ |
| 58 | 30 June 2015, 12:00:00 UTC | BSISO,Chan-hom | ○ | ○ | ○ | ○ |
| 59 | 1 July 2015, 00:00:00 UTC | BSISO,Chan-hom | ○ | ○ | ○ | ○ |
| 60 | 1 July 2015, 12:00:00 UTC | BSISO,Chan-hom | ○ | ○ | ○ | ○ |
| 61 | 13 August 2015, 12:00:00 UTC | | ○ | ○ | ○ | ○ |
| 62 | 14 August 2015, 00:00:00 UTC | ***Molave, Goni***, *Atsani* | ○ | ○ | ○ | ○ |
| 63 | 14 August 2015, 12:00:00 UTC | ***Molave, Goni***, *Atsani* | ○ | ○ | ○ | ○ |
| 64 | 15 August 2015, 00:00:00 UTC | ***Molave,*** Goni, Atsani | ○ | ○ | ○ | ○ |
| 65 | 15 August 2015, 12:00:00 UTC | ***Molave,*** Goni, Atsani | ○ | ○ | ○ | ○ |
| 66 | 16 August 2015, 00:00:00 UTC | ***Molave,*** Goni, Atsani | ○ | ○ | ○ | ○ |
| 67 | 16 August 2015, 12:00:00 UTC | ***Molave,*** Goni, Atsani | ○ | ○ | ○ | ○ |
| 68 | 17 August 2015, 00:00:00 UTC | ***Molave,*** Goni, Atsani | ○ | ○ | ○ | ○ |
| 69 | 17 August 2015, 12:00:00 UTC | ***Molave,*** Goni, Atsani | ○ | ○ | ○ | ○ |
| 70 | 18 August 2015, 00:00:00 UTC | ***Molave,*** Goni, Atsani | ○ | ○ | ○ | ○ |
| 71 | 18 August 2015, 12:00:00 UTC | Goni, Atsani | ○ | ○ | ○ | ○ |
| 72 | 19 August 2015, 00:00:00 UTC | Goni, Atsani | ○ | ○ | ○ | ○ |
| 73 | 19 August 2015, 12:00:00 UTC | Goni, Atsani | ○ | ○ | ○ | ○ |
| 74 | 20 August 2015, 00:00:00 UTC | Goni, Atsani | ○ | ○ | ○ | ○ |
| 75 | 20 August 2015, 12:00:00 UTC | Goni, Atsani | ○ | ○ | ○ | ○ |
| 76 | 21 August 2015, 00:00:00 UTC | Goni, Atsani | ○ | ○ | ○ | ○ |
| 77 | 6 September 2015, 00:00:00 UTC | Kilo, *Etau* | ○ | ○ | ○ | ○ |
| 78 | 6 September 2015, 12:00:00 UTC | Kilo, *Etau* | ○ | ○ | ○ | ○ |
| 79 | 7 September 2015, 00:00:00 UTC | Kilo, *Etau* | ○ | ○ | ○ | ○ |
| 80 | 7 September 2015, 12:00:00 UTC | Kilo, Etau | ○ | ○ | ○ | ○ |
| 81 | 8 September 2015, 00:00:00 UTC | Kilo, Etau | ○ | ○ | ○ | ○ |
| 82 | 8 September 2015, 12:00:00 UTC | Kilo, Etau | ○ | ○ | ○ | ○ |
| 83 | 9 September 2015, 00:00:00 UTC | Kilo, Etau | ○ | ○ | ○ | ○ |
| 84 | 9 September 2015, 12:00:00 UTC | Kilo, ***Etau*** | ○ | ○ | ○ | ○ |
| 85 | 10 September 2015, 00:00:00 UTC | Kilo, ***Etau*** | ○ | ○ | ○ | ○ |

(*2): run with the fixed version of MATSIRO (Sect. 2.2.3).

**Table 3. Output variables and domains.**

| Domain | Interval | Variable | Horizontal resolution |
|---|---|---|---|
| Global | 1 hour | Accumulated cloud ice (*cldi*), Accumulated cloud water (*cldw*), Outward longwave radiation (*olr*), Sea-level pressure (*psea*), 2-m specific humidity (*qs*), Sea surface temperature (*sst*), Total precipitable water (*tpw*), 2-m temperature (*ts*), 10-m zonal wind speed (*us*), 10-m meridional wind speed (*vs*) | 1.25° |
| | 1 hour (average) | Latent heat flux (*fllh*), Zonal wind stress (*flmu*), Meridional wind stress (*flmv*), Sensible heat flux (*flsh*), Precipitation (*prc*), Precipitation by cumulus parameterization (*prcc*) | 1.25° |
| | 3 hours | Cloud cover (*cvr*), Cloud water content (*cwc*), Cloud water (*qc* or *xc*), Cloud ice (*qi* or *xi*), rain water (*qr* or *xr*), snow (*qs* or *xs*), graupel (*qg* or *xg*), Specific humidity (*q*), Relative humidity (*rh*), Temperature (*t*), Zonal wind speed (*u*), Meridional wind speed (*v*), Vertical wind speed (*w*), Height (*z*) | 1.25° |
| | 3 hours (average) | Cumulus-induced heating (*hrcv*), Cloud-induced heating (*hrlc*), Radiation-induced heating (*hrr*), Turbulence-induced heating (*hrvd*), Cumulus-induced moistening (*qrcv*), Cloud-induced moistening (*qrlc*), Radiation-induced heating (*qrvd*), Cumulus-induced zonal acceleration (*urcv*), Turbulence-induced zonal acceleration (*urvd*), Cumulus-induced meridional acceleration (*vrcv*), Turbulence-induced meridional acceleration (*vrvd*) | 1.25° |
| Western North Pacific/Tropics | 1 hour | cldi, cldw, olr, psea, qs, sst, tpw, ts, us, vs | ~7 km |
| | 1 hour (average) | fllh, flmu, flmv, flsh, prc, prcc | ~7 km |
| | 3 hours | cvr, cwc, q, rh, t, u. v, w, z | ~7 km |
| | 3 hours (average) | hrcv, hrlc, hrr, hrvd, qrcv, qrlc, qrvd, urcv, urvd, vrcv, vrvd | ~7 km |

**Table 4. Brief description of the specifications for each global nonhydrostatic model.**

| | DFSM | GSM | MSSG | NICAM |
|---|---|---|---|---|
| Horizontal resolution | 7 km | 20 km | 7 km | 7 km |
| Horizontal Grid configuration | Reduced linear equally-spaced latitude grid | Reduced linear Gaussian grid | Yin-yang grid | Icosahedral grid |
| Number of grids in horizontal direction | 8845592 | 1312360 | 11184128 | 10485760 |
| Vertical coordinate | Hybrid sigma-pressure coordinate | Hybrid sigma-pressure coordinate | Terrain-following coordinate | Terrain-following coordinate |
| Vertical levels | 100 (top: 0.01 hPa, bottom: 999.0429 hPa (*3) (~8 m)) | 100 (top: 0.01 hPa, bottom: 999.0429 hPa (*3) (~8 m)) | 55 (top: 40 km, bottom: 75 m) | 38 (top: 36.7 km, bottom: 80 m) |
| Dynamical core | Nonhydrostatic spectral model using double Fourier series | Hydrostatic spectral model using spherical harmonics | Nonhydrostatic grid model using finite difference method | Nonhydrostatic grid model using finite volume method |
| Time step (s) | 200 | 400 | Variable | 30 |
| Cloud physics | Smith (1990) | Smith (1990) | Onishi and Takahashi (2012) | Tomita (2008) |
| Cumulus convection | Randall and Pan (1993) | Randall and Pan (1993) | Not used | Not used |
| Planetary boundary layer | MY2 (Mellor and Yamada, 1974, 1982) | MY2 (Mellor and Yamada, 1974, 1982) | MYNN2.5 (Nakanishi and Niino, 2009) | MYNN2 (Nakanishi and Niino, 2004; Noda et al., 2010) |
| Radiation | JMA (2013), Yabu (2013) | JMA (2013), Yabu (2013) | MstranX (Sekiguchi and Nakajima, 2008) | MstranX (Sekiguchi and Nakajima, 2008) |
| Land and ocean | SiB (JMA, 2013) | SiB (JMA, 2013) | Bucket Option: 3D ocean model | MATSIRO (Takata et al., 2003) Slab ocean model |
| Surface boundary layer | Louis (1982), Miller (1989, Ocean/ Unstable atmosphere) | Louis (1982), Miller (1989, Ocean/ Unstable atmosphere) | Zhang and Anthes (1982) for land surface; Fairall et al. (1996; 2003) for ocean surface | Louis (1979) |

(*3): Full-level pressure for surface pressure = 1000 hPa.

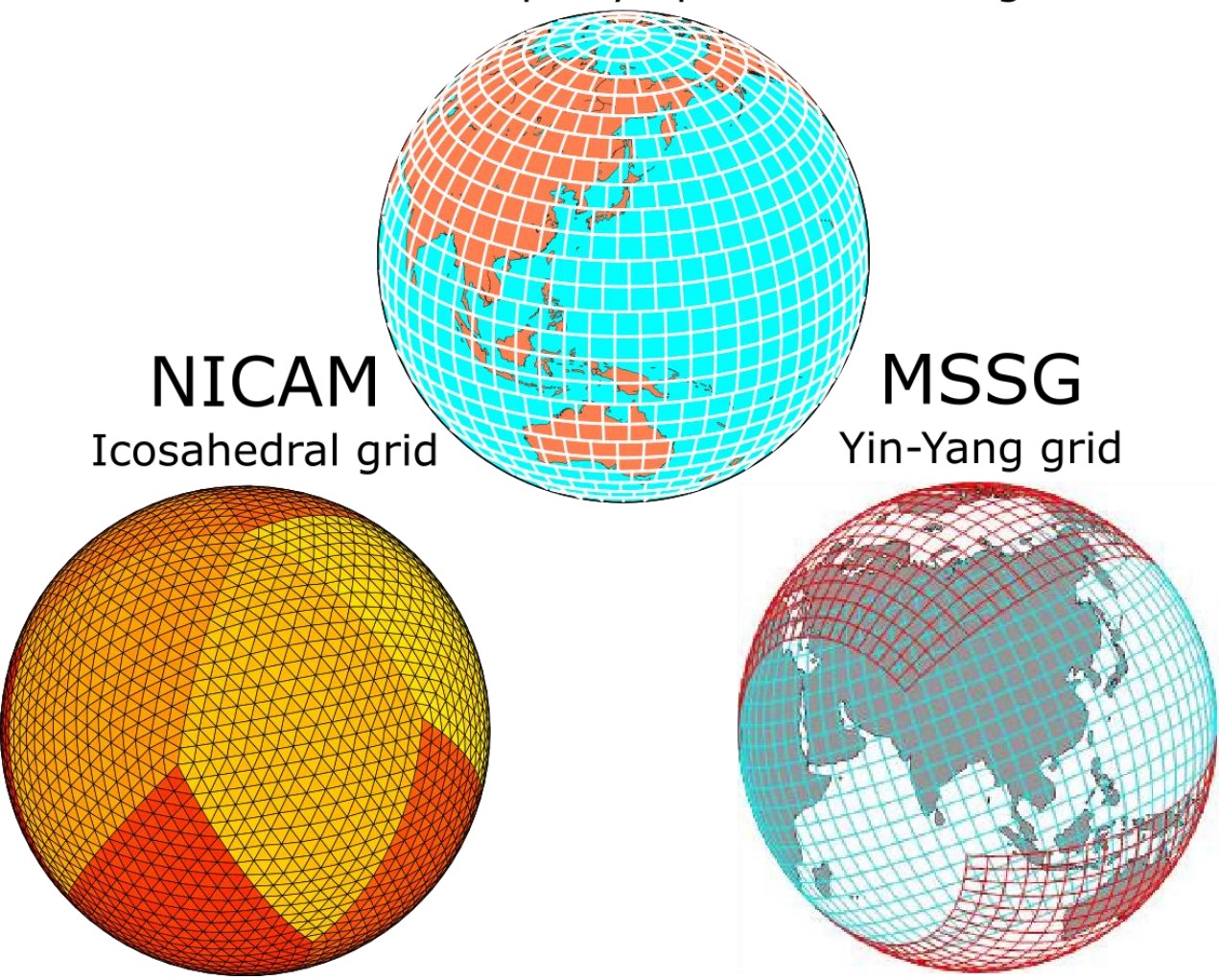

5    **Figure 1. Schematic diagram of the horizontal grid structures of the three models used in TYMIP-G7.**

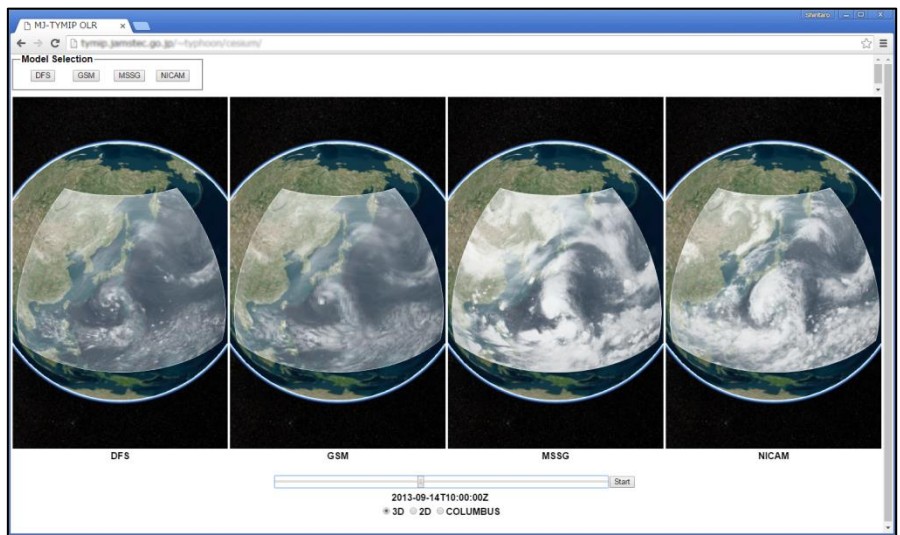

**Figure 2. Screen capture of the Web application: outgoing longwave radiation at 14 September 2013, 10:00:00 UTC simulated in experiments initialized at 12 September 2013, 06:00:00 UTC.**

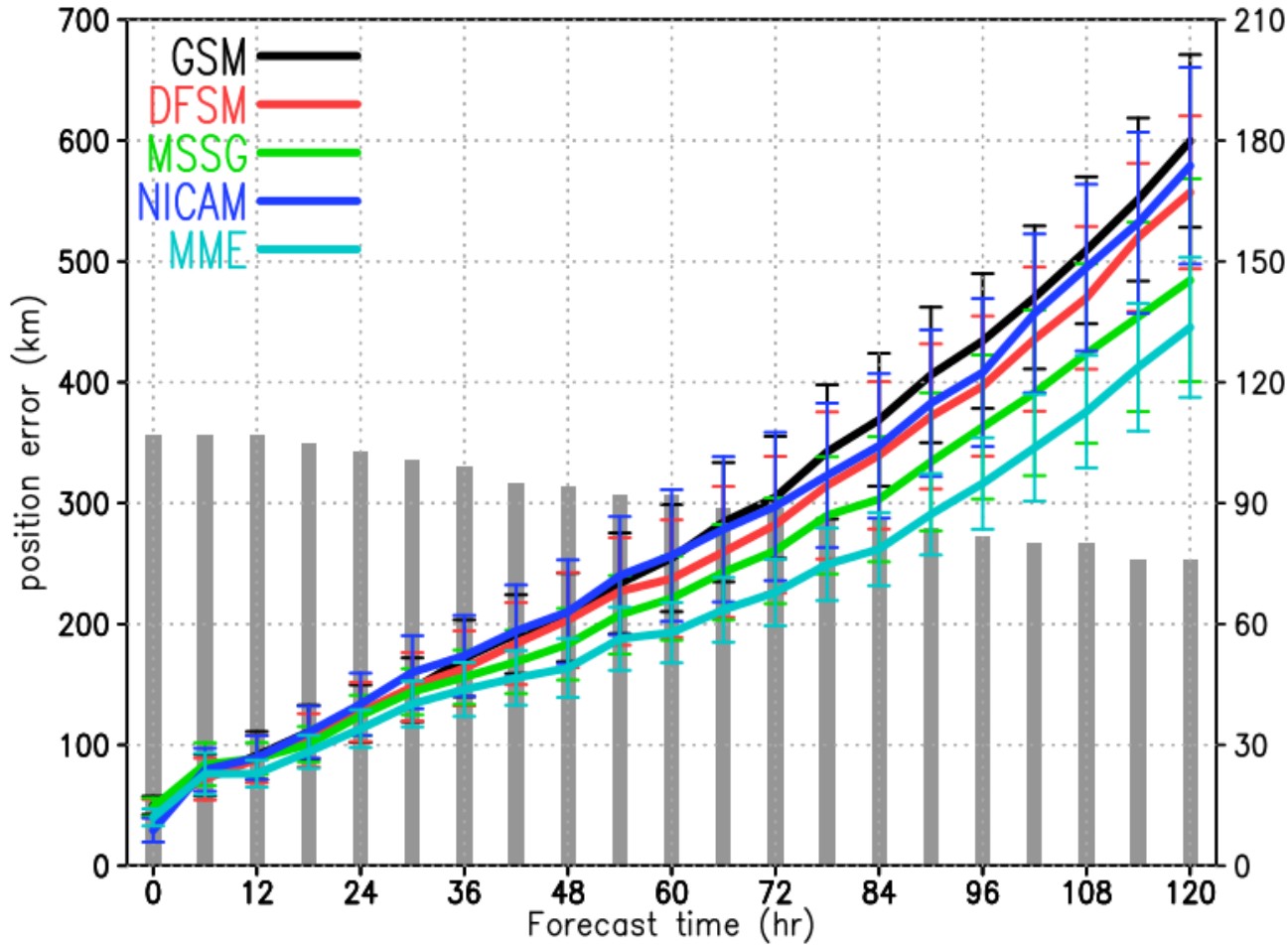

**Figure 3. Errors in the track prediction for GSM, DFSM, MSSG, NICAM and MME (in the second stage). Each grey bar indicates the number of samples at each forecast time (right-vertical axis). Error bars indicate 95% confidence levels of the central pressure difference between the prediction and the RSMC Tokyo best-track data.**

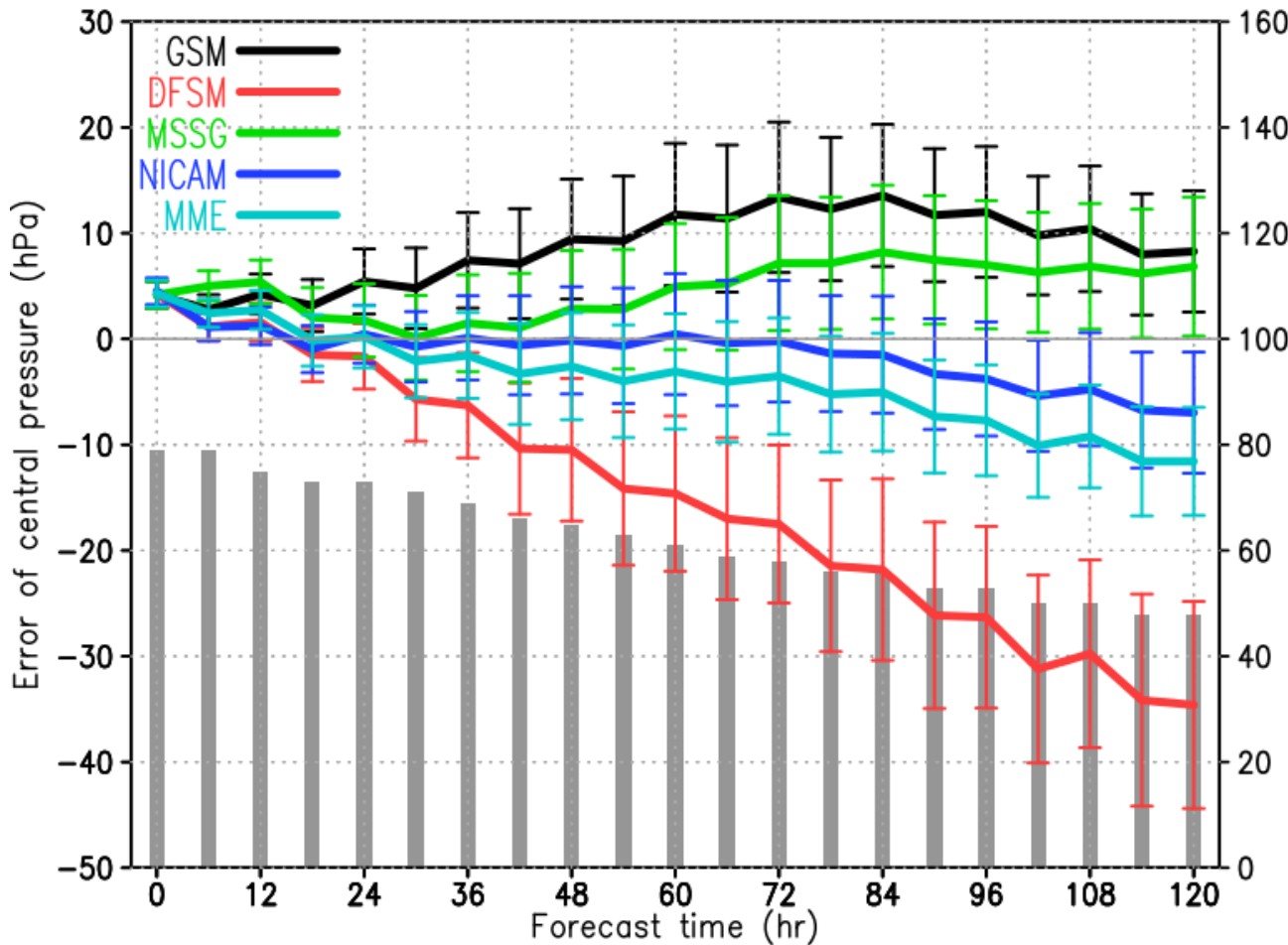

**Figure 4. Errors in the predictions of the central pressure for GSM, DFSM, MSSG, NICAM and MME (in the second stage). Each grey bar indicates the number of samples at each forecast time (right-vertical axis). Error bars indicate 95% confidence levels of the central pressure difference between the prediction and the RSMC Tokyo best-track data.**

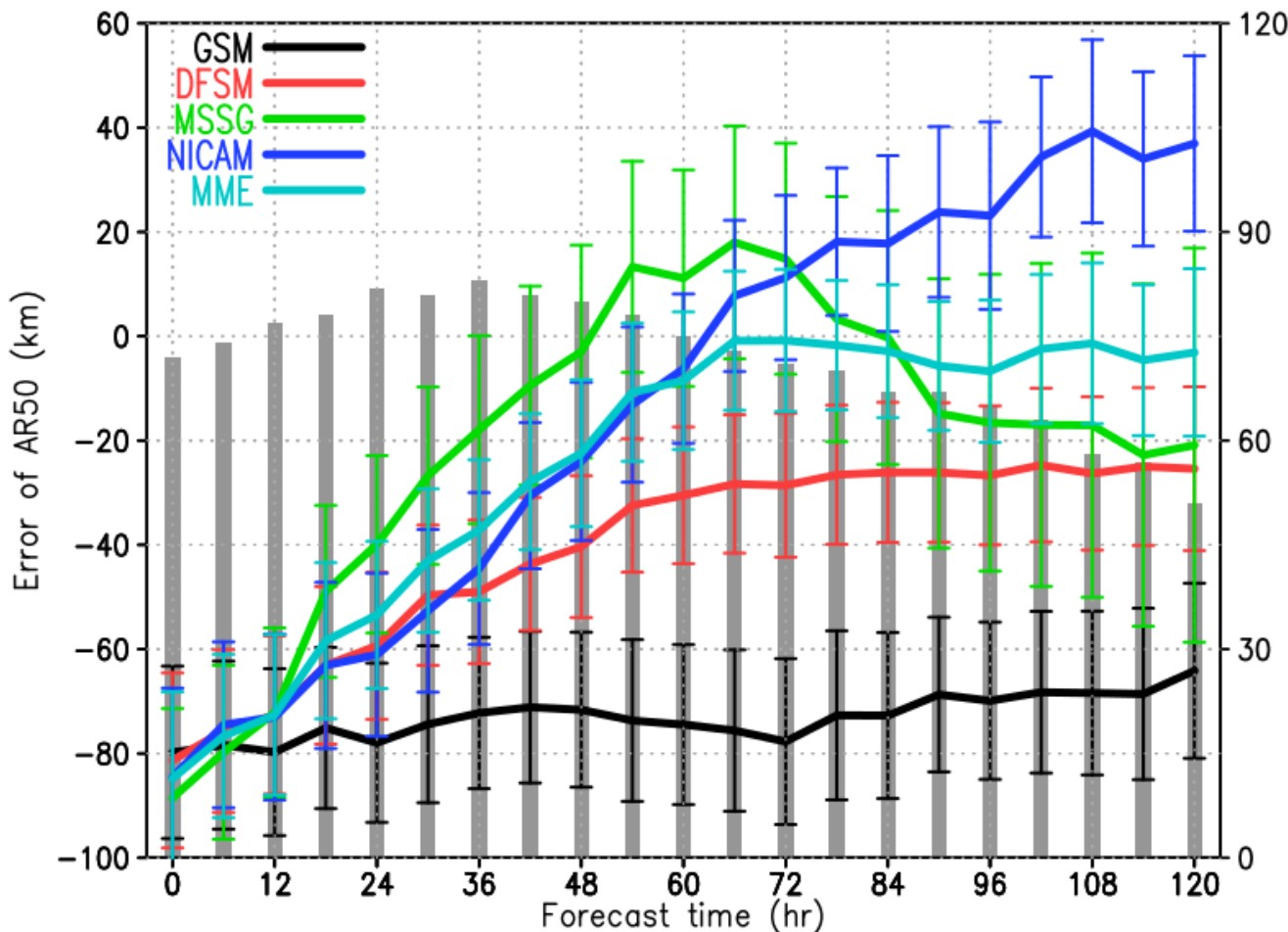

**Figure 5. Errors in the averaged radius of the 50-knot wind (AR50) for GSM, DFSM, MSSG, NICAM and MME (in the second stage). Each grey bar indicates the number of samples at each forecast time (right-vertical axis). Error bars indicate 95% confidence levels of the AR50 difference between the prediction and the RSMC Tokyo best-track data.**

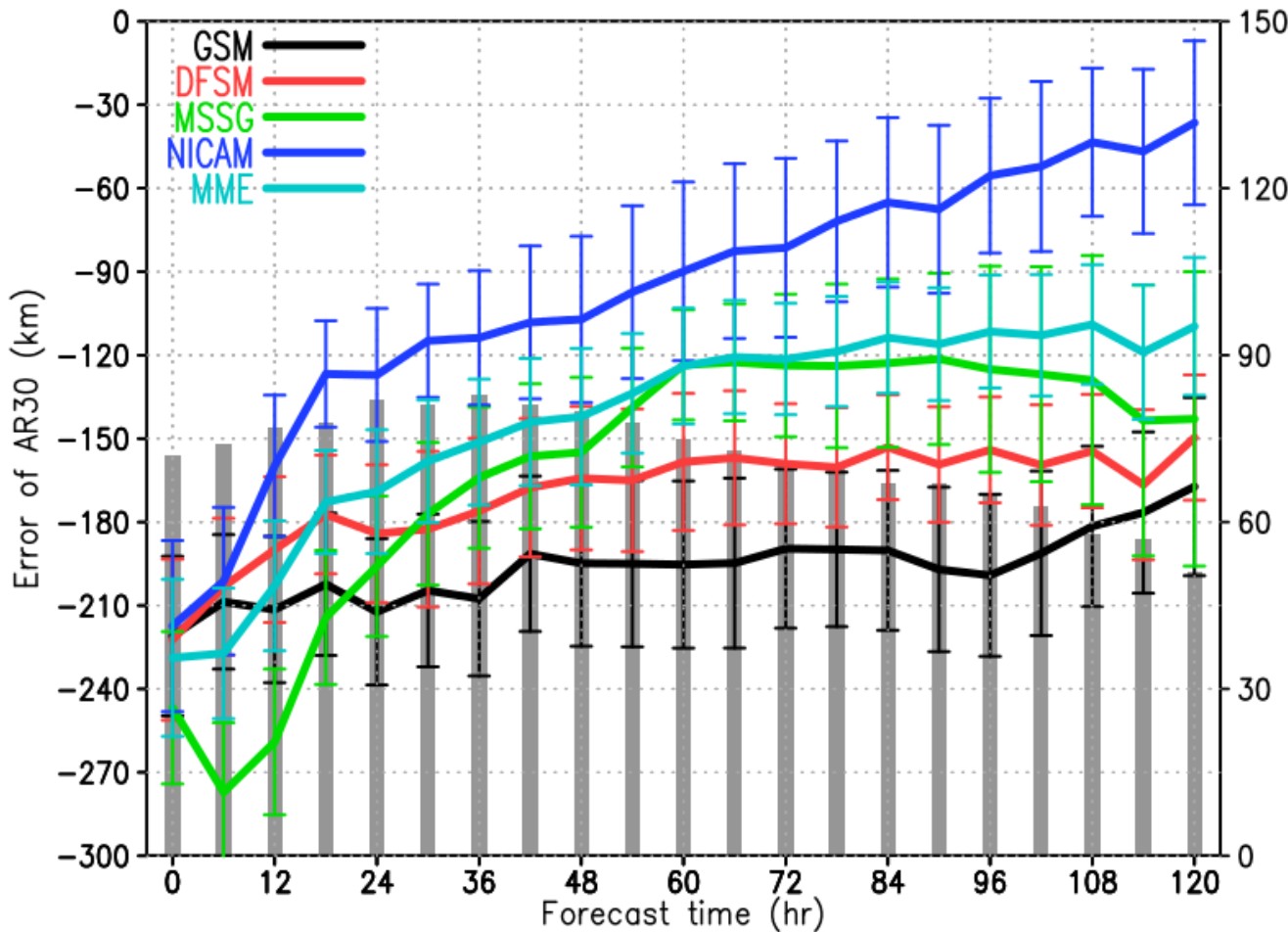

**Figure 6. Errors in the averaged radius of the 30-knot wind (AR30) for GSM, DFSM, MSSG, NICAM and MME (in the second stage). Each grey bar indicates the number of samples at each forecast time (right-vertical axis). Error bars indicate 95% confidence levels of the AR30 difference between the prediction and the RSMC Tokyo best-track data.**

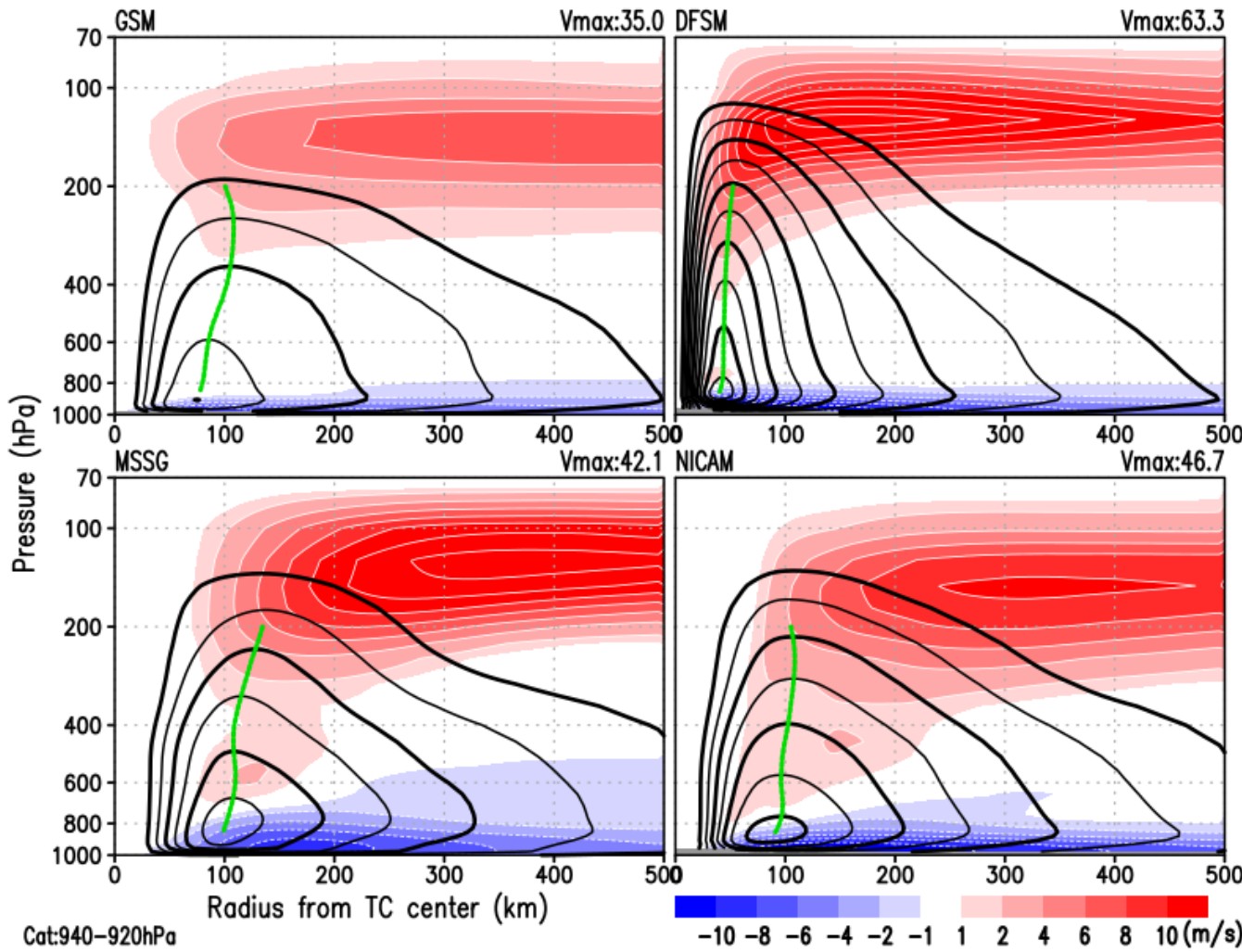

**Figure 7. Composite analysis of the radius–height cross section of the axisymmetric mean radial (shaded) and tangential (contour) wind speed for TCs at the time of the analysed central pressure between 920–940 hPa in the RSMC Tokyo best-track data. Contour intervals are 5 m s⁻¹ (values > 15 m s⁻¹ are plotted). The green line depicts the RMW between 850 hPa and 200 hPa. The grey shading at the bottom of each panel is below the surface.**