# Peer review of "Global 7-km mesh nonhydrostatic Model Intercomparison Project for improving TYphoon forecast (TYMIP-G7): Experimental design and preliminary results"

_Geoscientific Model Development, 2016_

## Referee Comment (RC1) · Anonymous Referee #1 · 16 Aug 2016

This manuscript details the performance of a suite of models within the TYMIP-G7 project. These models aim to assess the performance of very high resolution global setups (three 7km models, one 20km model) when forecasting typhoons (tropical cyclones, TCs) in the western Pacific. The authors demonstrate that the 7km models perform better over their sample set, both in terms of forecasting mean TC track and intensity, as well as potentially resolving asymmetric features more accurately than the 20km model.

The resolutions utilized in this study are some of the finer grid spacings applied to

global TC forecasting and represent an important contribution within the R2O framework given that operational models will likely begin to be integrated in this resolution space within the next decade. That said, the goals of the manuscript are somewhat unclear. Is the manuscript purely describing the TYMIP-G7 project framework such that it can be referenced in future studies? Or do the authors seek to describe fundamental differences in model results and attribute them to different model configurations? The authors bounce back and forth a bit between the two and the analysis of TC forecasts (beyond mean statistics) is somewhat weak, particularly in the final quarter of the manuscript. The result that increasing resolution improves TC forecasts is not tremendously novel in the community. The analysis of the different structure of the forecasted TCs is intriguing (although require significantly more work in future manuscripts) but the authors only select a particular forecast cycle to perform analysis on, which seems tenuous (at best) given the spread in TC intensity forecasts discussed earlier in the manuscript.

My recommendation is for "major revisions." I think the authors would be well-served to tighten up the description of the simulations and model configurations, which would give a very clear citation for future papers using TYMIP-G7 data. In addition, while the authors do not need to explain why models perform differently (those are additional projects in and of themselves), it would be useful to have something more than a single forecast initialization analyzed, particularly for Figs. 6 and 8. My preference would be to present a mean structure over multiple forecast cycles and explain that these differences exist in these model configurations and require additional analysis in the future. I have elaborated on major and minor critiques below.

In addition, there are phrasings that are somewhat awkward and grammatically incorrect for an English journal. I have noted some below but it is not meant to be an exhaustive list. My recommendation would be for a native English speaker to proofread this manuscript thoroughly before resubmission.

Major comments:

- There is very little that can be said about model differences based on single forecast experiments. While I am aware that this manuscript is not intended to explain all of the physical differences (of which there might be many, particularly within the subgrid parameterization suites), I am worried that there is little utility in Fig. 6 and 8. I would anticipate being able to find cases where, for example, TCs have more asymmetric structure in GSM (even with lower resolution) or look more like observations, due to the fact that there are many forecast cycles from which to pick from. The same goes for the depth and structure of the axisymmetric circulation. Picking single members from the envelope of Fig. 4 implies that you cannot adequately understand model differences because you aren't removing run-to-run variability. In Fig. 8, it's possible that the NICAM signal (TC with lower outflow jet and shallower inflow) is a physical signal (perhaps due to the NICAM setup itself) but it also may be that that particular forecast in NICAM had more vertical wind shear than the other model configurations. My preference here would be for there to be either multiple TCs explored or perhaps some sort of average across a number of forecast cycles (say, Fig. 8 could be the average of 20 different TCs at +96 hour lead time from 20 different forecast initializations).

- Why is only the second stage shown in Fig. 3 but both stages are included in Fig. 4? This is especially relevant since the authors state that "track errors in MSSG were larger than those of GSM" in Stage 1, which is the opposite of the Stage 2 results (Fig. 3). If the errors associated with precipitable water (Page 8, line 25) were severe enough to eliminate their usage in Fig. 3, why weren't they eliminated in Fig. 4? Also, why are there error bars in Fig. 4 but not in Fig. 3? Error bars should be included in Fig. 3 to give a sense as to the spread around the mean. It is difficult to understand whether those differences in track are "significant" (in either a statistical sense or just by subjectively assessing the figure).

- The timing results are very underdeveloped. For example, what is "execution efficiency?" To be honest, I'm not sure if this adds a great deal to the manuscript. Timing studies seem most useful either a) when as many variables are constrained as possi-

СЗ

ble (i.e., same resolution, different physics, etc.) or b) operationally, when a wall clock time benchmark threshold is required. For example, here DFSM is much faster, so in an operational sense, a forecaster might say "why don't we just use DFSM?" However, a more rigorous timing analysis might want to demonstrate the strong and weak scaling properties of the model and what happens if different subgrid parameterizations are used. Furthermore, Table 5 currently investigates only one forecast cycle. Individual forecasts may have different timings (even with the same model) for a variety of reasons (different load on the computing cluster, how the communication is spread amongst nodes, failures/bottlenecks during I/O write to disks, etc.). My recommendation would be to just remove the table (since this is R2O) and spend a brief paragraph discussing mean timings (i.e., over multiple forecast cycles), but emphasizing that there are many, many different aspects of each model configuration that lead to the disparate timings.

- The authors mention "errors" in Stage 1 forecasts multiple times during the manuscript but don't elaborate significantly. My preference would be for any changes/corrections between Stage 1 and Stage 2 that persist in the data to be noted clearly such that future analysis with TYMIP-G7 data can refer back to it (note, that if the authors corrected these issues and merely re-ran Stage 1 with the updated settings, there is no reason to mention this as long as the "incorrect" Stage 1 data is overwritten).

Minor comments:

- Page 2, Line 10: '... is to avoid that transform.' Please cite a reference.

- Page 10, Line 11: '... Skamarock (2004) stated that seven times...' this is dependent on the numerical scheme and not universal across all models. See Kent et al., 2014, JCP.

- Page 10: Line 28: Is there anything in this manuscript that evaluates rapid intensification (RI) critically? A figure such as Fig. 5 could be useful, but if Delta\_SLP (change in surface pressure) is calculated, not absolute surface pressure. Otherwise, RI seems neglected, so I wouldn't include this as a main result.

- Fig. 4., it appears the initialization is too weak across all models ( $\sim$ 5 hPa), which could propagate through the intensity forecasts at long leads. This is particularly relevant for the DFSM model which is initialized too weak yet develops TCs that are generally too strong.

Grammar/Typos:

- Page 2, Line 27: '... form on annual average in the western North Pacific...' is awkward. Could be 'Since an average of 26 TCs (XXXX) form on average in the western North Pacific....'

- Page 2, Line 39: 'to' should be 'too'
- Page 3, Line 37: '... diurnal cyclone...' should be '... diurnal cycle...'
- Page 4, Line 5: '... most activate ...' should be '... most active ...'

- Page 9, Line 32-33: 'However, precipitation patterns...' should be 'However, the precipitation patters...'

---

## Referee Comment (RC2) · Anonymous Referee #2 · 11 Oct 2016

This paper provides a preliminary assessment of tropical cyclone quality obtained from three high-resolution models. Although the paper is well-written, understandable and provides interesting results, I am somewhat concerned about framing the paper as a model intercomparison. Namely, I would expect a paper that lays out an intercomparison effort would provide substantive details on how one can quantify success. This is particularly evident through section 5.3, where the assessment is purely qualitative and, by avoiding any negativity about particular modeling systems, fails to call out what seem to be deficiencies in the structure of the typhoon that arises in the MSSG and DFSM models. To address this concern, it is suggested that the authors explicitly call

out the metrics (computational performance, track error, intensity error, structural error) that could be used to quantitatively assess model performance, along with how one could quantify success under these metrics. This listing is analogous to section 4.1, but with further quantification of success or error under each criteria. Further, a tabulated intercomparison of the compared models that shows successes / deficiencies would be helpful to the reader to more clearly see how they intercompare.

Some additional comments are given below:

Page 6, line 6: Why is the standalone MSSG-A used when a coupled ocean-atmosphere version is available?

Page 7, line 11: Does the slab ocean model react appropriately to the passing TC by generating a cold wake?

Page 7, line 4-5: "split-explicit" and "horizontally explicit and vertically implicit" typically refer to different techniques. The former uses explicit sub-cycling to deal with some vertically propagating wave modes, whereas the latter uses an implicit solve for all vertical terms.

Page 9, line 27: Is the appearance of the "wavy" structure of the tropical cyclone associated with Gibbs' oscillations that arise from the spectral-transform nature in the model?

Page 9, line 35-37: Some additional discussion should be provided here regarding correctness. It seems that NICAM is the only one that produces a structure that matches observations – is that a correct assessment? Since the paper advocates for this hindcast strategy for assessing model quality, it should be more explicit on how one can actually evaluate the models using a mechanism that is not purely qualitative.

Page 10, line 7: Again, what is "correct"? The focus on qualitative model differences in this paragraph gives no insight into actually evaluations of the model.

Page 10, line 11: This is actually highly dependent on the numerical methods employed

in the dynamical core and associated diffusion scheme. For a model like DFSM one would expect a finest resolved dynamical mode closer to 4dx, whereas for NICAM, which uses a co-located finite volume method, one would expect a finest resolved mode closer to 12dx. See Ullrich (2014).

Page 10, line 17: Again, additional information is needed on model correctness.

Page 11, line 13: It is stated earlier that NICAM is used in coupled atmosphere-ocean mode, so this statement is not quite correct.
* * *

---

## Author Response (AR1)

Reply to reviewer 1

We would like to appreciate the valuable comments which help for improving the manuscript. In the revision, we clarify the metrics to quantitatively evaluate model performance and add some validation results concerning TC surface wind structure (Figures 5 and 6). Figures 1 and 2 are redrawn by using the result of Stage 2 only with 95% confidence levels as error bars rather than standard deviation. Careful analysis of simulated TC position revealed that some misdetection occurred for very weak TC cases. These cases are excluded from validation.

Point-to-point response are following.

*That said, the goals of the manuscript are somewhat unclear.*
*Is the manuscript purely describing the TYMIP-G7 project framework such that*
*it can be referenced in future studies? Or do the authors seek to describe fundamental*
*differences in model results and attribute them to different model configurations?*
*The authors bounce back and forth a bit between the two and the analysis of TC forecasts*
*(beyond mean statistics) is somewhat weak, particularly in the final quarter of the*
*manuscript. The result that increasing resolution improves TC forecasts is not tremendously*
*novel in the community. The analysis of the different structure of the forecasted*
*TCs is intriguing (although require significantly more work in future manuscripts) but*
*the authors only select a particular forecast cycle to perform analysis on, which seems*
*tenuous (at best) given the spread in TC intensity forecasts discussed earlier in the*
*manuscript.*
*My recommendation is for "major revisions." I think the authors would be well-served*
*to tighten up the description of the simulations and model configurations, which would*
*give a very clear citation for future papers using TYMIP-G7 data. In addition, while*
*the authors do not need to explain why models perform differently (those are additional*
*projects in and of themselves), it would be useful to have something more than a single*
*forecast initialization analyzed, particularly for Figs. 6 and 8. My preference would*
*be to present a mean structure over multiple forecast cycles and explain that these*
*differences exist in these model configurations and require additional analysis in the*
*future. I have elaborated on major and minor critiques below.*

In this revision, we clearly state that the aim of the manuscript at the last of Section 1: describing the specification of TYMIP-G7 and a set of metrics, and showing results concerning the metrics. We deleted Figs.6 and 8 which showed simulated TC structure for a specific case, but added a composite of axisymmetric primary and secondary circulation at the mature phase of TC to discuss the difference in simulated structure of TCs. Yes, we need further works to make clear what causes

difference in simulated TC structure. Thank you for the comment.

*In addition, there are phrasings that are somewhat awkward and grammatically incorrect for an English journal. I have noted some below but it is not meant to be an exhaustive list. My recommendation would be for a native English speaker to proofread this manuscript thoroughly before resubmission.*

We are so sorry for our English quality. We ordered an English editing service by Enago before resubmission.

*There is very little that can be said about model differences based on single forecast experiments. While I am aware that this manuscript is not intended to explain all of the physical differences (of which there might be many, particularly within the subgrid parameterization suites), I am worried that there is little utility in Fig. 6 and 8. I would anticipate being able to find cases where, for example, TCs have more asymmetric structure in GSM (even with lower resolution) or look more like observations, due to the fact that there are many forecast cycles from which to pick from. The same goes for the depth and structure of the axisymmetric circulation. Picking single members from the envelope of Fig. 4 implies that you cannot adequately understand model differences because you aren't removing run-to-run variability. In Fig. 8, it's possible that the NICAM signal (TC with lower outflow jet and shallower inflow) is a physical signal (perhaps due to the NICAM setup itself) but it also may be that that particular forecast in NICAM had more vertical wind shear than the other model configurations. My preference here would be for there to be either multiple TCs explored or perhaps some sort of average across a number of forecast cycles (say, Fig. 8 could be the average of 20 different TCs at +96 hour lead time from 20 different forecast initializations).*

Thank you for the comment. It should be very useful for further detailed analyses. In the revision, we deleted Figs.6 and 8 which showed simulated TC structure for a specific case, but added a composite of axisymmetric primary and secondary circulation at the mature phase of TC.

*Why is only the second stage shown in Fig. 3 but both stages are included in Fig. 4? This is especially relevant since the authors state that "track errors in MSSG were larger than those of GSM" in Stage 1, which is the opposite of the Stage 2 results (Fig. 3). If the errors associated with precipitable water (Page 8, line 25) were severe enough to eliminate their usage in Fig. 3, why weren't they eliminated in Fig. 4? Also,*

*why are there error bars in Fig. 4 but not in Fig. 3? Error bars should be included in Fig. 3 to give a sense as to the spread around the mean. It is difficult to understand whether those differences in track are "significant" (in either a statistical sense or just by subjectively assessing the figure).*

We used the result of Stage 2 only for Figs. 3 and 4 and added error bars showing 95% confidence levels rather than one standard deviation.

*The timing results are very underdeveloped. For example, what is "execution efficiency?" To be honest, I'm not sure if this adds a great deal to the manuscript. Timing studies seem most useful either a) when as many variables are constrained as possi-ble (i.e., same resolution, different physics, etc.) or b) operationally, when a wall clock time benchmark threshold is required. For example, here DFSM is much faster, so in an operational sense, a forecaster might say "why don't we just use DFSM?" However, a more rigorous timing analysis might want to demonstrate the strong and weak scaling properties of the model and what happens if different subgrid parameterizations are used. Furthermore, Table 5 currently investigates only one forecast cycle. Individual forecasts may have different timings (even with the same model) for a variety of reasons (different load on the computing cluster, how the communication is spread amongst nodes, failures/bottlenecks during I/O write to disks, etc.). My recommendation would be to just remove the table (since this is R2O) and spend a brief paragraph discussing mean timings (i.e., over multiple forecast cycles), but emphasizing that there are many, many different aspects of each model configuration that lead to the disparate timings.*

Thank you for the comment. In the revision, we used computational resource for a 5-day forecast (node-hours) as a metrics to evaluate the timing of the model. The amount of resource is hardly variable among cases because computational nodes are occupied for a model experiment and disk I/O is performed from/to the work disk mounted on each computational node. We also discussed many aspects which affect timings.

*The authors mention "errors" in Stage 1 forecasts multiple times during the manuscript but don't elaborate significantly. My preference would be for any changes/corrections between Stage 1 and Stage 2 that persist in the data to be noted clearly such that future analysis with TYMIP-G7 data can refer back to it (note, that if the authors corrected these issues and merely re-ran Stage 1 with the updated settings, there is no reason*

*to mention this as long as the "incorrect" Stage 1 data is overwritten).*

We decided to rerun the experiments in Stage 1 by MSSG using this year's budget but they have not completed yet. Because Stage 2 has enough samples to examine difference in TC track, intensity, and structure, we used Stage 2 throughout the revision. Since we believe describing causes of failure in MSSG would help some model developers, we remained the description. Thank you for your understandings.

*- Page 2, Line 10: '... is to avoid that transform.' Please cite a reference.*
*- Page 10, Line 11: '... Skamarock (2004) stated that seven times...' this is dependent*
*on the numerical scheme and not universal across all models. See Kent et al., 2014,*
*JCP.*

Deleted.

*- Page 10: Line 28: Is there anything in this manuscript that evaluates rapid intensification*
*(RI) critically? A figure such as Fig. 5 could be useful, but if Delta_SLP (change*
*in surface pressure) is calculated, not absolute surface pressure. Otherwise, RI seems*
*neglected, so I wouldn't include this as a main result.*

We deleted some sentences concerning RI.

*- Fig. 4., it appears the initialization is too weak across all models (~5 hPa), which could*
*propagate through the intensity forecasts at long leads. This is particularly relevant for*
*the DFSM model which is initialized too weak yet develops TCs that are generally too*
*strong.*

Thank you for the comment. Initial bias of TC central pressure is mentioned in the revision.

*Grammar/Typos:*
*- Page 2, Line 27: '... form on annual average in the western North Pacific...' is awkward.*
*Could be 'Since an average of 26 TCs (XXXX) form on average in the western North Pacific....'*
*- Page 2, Line 39: 'to' should be 'too'*
*- Page 3, Line 37: '... diurnal cyclone...' should be '... diurnal cycle...'*
*- Page 4, Line 5: '... most activate...' should be '... most active...'*
*- Page 9, Line 32-33: 'However, precipitation patterns...' should be 'However, the precipitation*

*patters...'*

All comments concerning above grammar and spelling errors are corrected in the revision. Thank you for your careful review.

Reply to reviewer 2

We would like to appreciate the valuable comments which help for improving the manuscript.

In the revision, we clarify the metrics to quantitatively evaluate model performance and add some validation results concerning TC surface wind structure (Figures 5 and 6). Figures 1 and 2 are redrawn by using the result of Stage 2 only with 95% confidence levels as error bars rather than standard deviation. Careful analysis of simulated TC position revealed that some misdetection occurred for very weak TC cases. These cases are excluded from validation.

Point-to-point response are following.

This paper provides a preliminary assessment of tropical cyclone quality obtained from three high-resolution models. Although the paper is well-written, understandable and provides interesting results, I am somewhat concerned about framing the paper as a model intercomparison. Namely, I would expect a paper that lays out an intercomparison effort would provide substantive details on how one can quantify success. This is particularly evident through section 5.3, where the assessment is purely qualitative and, by avoiding any negativity about particular modeling systems, fails to call out what seem to be deficiencies in the structure of the typhoon that arises in the MSSG and DFSM models. To address this concern, it is suggested that the authors explicitly callout the metrics (computational performance, track error, intensity error, structural error) that could be used to quantitatively assess model performance, along with how one could quantify success under these metrics. This listing is analogous to section 4.1, but with further quantification of success or error under each criteria. Further, a tabulated intercomparison of the compared models that shows successes / deficiencies would be helpful to the reader to more clearly see how they intercompare.

In this revision, we clarified metrics to quantitatively evaluate model performance (Section 4.1) and the evaluation results (Section 5). Thank you for the valuable comment.

Page 6, line 6: Why is the standalone MSSG-A used when a coupled oceanatmosphere version is available?

Because the other models are not coupled with full 3D ocean model, the standalone MSSG-A was used so far. As we discussed in Section 6, we would like to examine the ocean effect using AO coupled MSSG as well as AO coupled NICAM. Thank you for the understanding.

Page 7, line 11: Does the slab ocean model react appropriately to the passing TC by

generating a cold wake?

The slab ocean model just calculates local heat budget between atmosphere and ocean slab. Therefore, no cooling due to vertical mixing or Ekman pumping, but cooling by shielding effect of short wave by clouds occurs (Page 7, lines 17-19).

Page 7, line 4-5: "split-explicit" and "horizontally explicit and vertically implicit" typically refer to different techniques. The former uses explicit sub-cycling to deal with some vertically propagating wave modes, whereas the latter uses an implicit solve for all vertical terms.

Thank you for the comment. This part has been corrected in the revision (Page 7, line 11).

Page 9, line 27: Is the appearance of the "wavy" structure of the tropical cyclone associated with Gibbs' oscillations that arise from the spectral-transform nature in the model?

Figure 6 was deleted in the revision.

Page 9, line 35-37: Some additional discussion should be provided here regarding correctness. It seems that NICAM is the only one that produces a structure that matches observations – is that a correct assessment? Since the paper advocates for this hindcast strategy for assessing model quality, it should be more explicit on how one can actually evaluate the models using a mechanism that is not purely qualitative.
Page 10, line 7: Again, what is "correct"? The focus on qualitative model differences in this paragraph gives no insight into actually evaluations of the model.
Page 10, line 17: Again, additional information is needed on model correctness.

In this revision, we clarified metrics to quantitatively evaluate model performance (Section 4.1) and the evaluation results (Section 5). Thank you for the valuable comment.

Page 10, line 11: This is actually highly dependent on the numerical methods employed in the dynamical core and associated diffusion scheme. For a model like DFSM one would expect a finest resolved dynamical mode closer to 4dx, whereas for NICAM, which uses a co-located finite volume method, one would expect a finest resolved mode closer to 12dx. See Ullrich (2014).

Thank you for the comment. This part is deleted in the revision.

Page 11, line 13: It is stated earlier that NICAM is used in coupled atmosphere-ocean mode, so this statement is not quite correct.

The slab ocean does not calculate any ocean dynamics like vertical mixing and advection (Page 7, lines 17-19). Here, we would like to state that coupling with a 3D full ocean model is needed to examine impact of ocean cooling by TC. The statement is modified in the revision (Page 11, line 25).

[revised manuscript text omitted]
 set of metrics to evaluate the regarding TC forecast by global models to validate performance. metrics of TC forecast by global models as follows:

(1)  ComputationalC resources for a 5-day forecast on the Earth simulator (node-hours)

(2)  TC track (position) error every 6 hours of forecast time along with forecast time (km)

(3)  TC intensity (central pressure) error every 6 hours of forecast time along with forecast time (hPa)

(4)  Averaged radius of surface 50-knot (25 m s$^{-1}$) wind (AR50) error (km)

(5)  Averaged radius of surface 30-knot (15 m s$^{-1}$) wind (AR30) error (km).

It is important for the operational model that the calculation is completed in less time with smaller computational resources so that we applied the metric (1). A demand for smaller amount of computational resources is important for an operational model. This is considered by (1). The metrics (2)–(5) measuresmeasure the accuracy of the track, intensity, and surface wind structure prediction based on the and these can be evaluated using RSMC Tokyo best-track data.

**4.21 TC tracking**

We extract TC tracking tracks in each from the model experiments using the hourly mean sea level pressure (SLP) data with a horizontal resolution of ~7 km for DFSM, MSSG and NICAM, and 20 km for GSM. A TC centre is defined as a minimum SLP point from the predicted mean SLP field smoothed 100 times by a 1-2-1 filter, for each longitude and latitude. The initial TC centre is defined within a radius of 1° from a centre position based on the RSMC Tokyo best-track data. The next centre position is defined as the minimum SLP point from the smoothed mean SLP field within a radius of 1° from the previous centre position. The tracking terminates when the minimum SLP points reach a a proximityproximity of 1° from the lateral boundary in the domain of the output data. The tracking algorithm works well for almostnearly all cases, but; however, misdetection occurred for some very weak TCs. These cases were excluded from the validation.

**4.3 Calculation method of AR50 and AR30**

The RSMC Tokyo best-track data contains the longest and shortest radii of 50-knot and 30-knot wind speedspeeds and itstheir direction. Observed AR50 and AR30 are defined as the average of the longest and shortest radii of the 50-knot and 30-knot wind speedspeeds, respectively. The directions of the longest and shortest radii are defined by eight directions

(N, NE, E, SE, S, SW, W, and NW) in the best-track data. Therefore, we calculated the radii of the 50-knot and 30-knot wind in the model in each of the eight directions first, and then determined the direction of the longest and shortest radii. Then, the radii in those two directions were averaged to obtain AR50 and AR30.

**4.4 Multi-model ensemble mean**

The multi-model ensemble mean (MME) is applied to the three 7-km mesh models (DFSM, MSSG, and NICAM). MME is a simple ensemble average derived from a combination of individual models, which reduces the average forecast error relative to the best individual predictions by the individual models. MME also provides additional information about the forecast uncertainty, enhancing forecast confidence (Goerss, 2000; Yamaguchi et al., 2012).

**4.5 Visualization**

We have developed a Web application that allows the simultaneous visualization of multi-model results. Figure 2 shows a screen capture of this application, which portrays digital globes using Cesium.js (Analytical Graphics, Inc., 2015), a WebGL-based virtual globe and map engine. Visualization results of each model are overlaid on them. We used Volume Data Visualizer for Google Earth (VDVGE; Kawahara, 2012; Kawahara, 2015) to depict visualization results for the overlay. VDVGE is a visualization software that exports visualization results in the KML format, a data format suitable for Google Earth. An option to export in the CZML format, suitable for Cesium.js, has recently been implemented in VDVGE. The present Web application enables us to view the animation display for time-series visualization results of each model, while synchronously changing the three-dimensional viewpoint. An option to display each model result selectively is also available. This application enables the four models to be easily compared.

**5 Results**

**5.1 Computational resources**

Computational performance is an important metric for an operational numerical weather forecast model. DFSM, MSSG, and NICAM models consumed computational resources equivalent to 682, 2330, and 1155 node-hours, respectively for a case on 12 September 2013, 00:00:00 UTC. These quantities did not vary greatly between cases because the computational nodes were occupied in each calculation and the disk I/O was executed from/to the work disk mounted on each computational node. Note that the computational resources required for each model are highly dependent on the model specifications of the Earth Simulator (e.g., the physics scheme, advection scheme, number of vertical layers, vertical resolution, and time

step) and the degree of optimization for the Earth Simulator. ~~computational resources needed for each model are highly depend on model specification (e.g., physics schemes, advection schemes, number of vertical layers, vertical resolutions, and time steps) and degree of optimization for the earth simulator. For example, DFSM was optimized for the Earth Simulator of the project, which is about four times faster than beforethe conventional one. Among the 7-km mesh models, DFSM requires the least computational resources, particularly in node hours. This is largely because of the relatively long time step (Table 3), owing to the semi-implicit semi-Lagrangian scheme within that model. More sophisticated cloud microphysics schemes in MSSG and NICAM than that in DFSM is also a factor in increased computational cost of the first two models. MSSG requires the greatest computational resource, about twice that of NICAM, even though both models used the conventional advection schemes. The difference in node hours between MSSG and NICAM are mainly attributable to the difference in vertical resolutions and number of vertical levels, which are sensitive to the time step setting.~~

**5.2 Track predictions**

To quantify the advantage of using finer resolution on TC track prediction, we examined the time series of  TC track prediction errors with reference to the RSMC Tokyo best track for the second stage (Figure 3).

~~In the first stage, TC track prediction by DFSM and NICAM showed better performance after forecast times (FTs) of 36 h and 96 h, respectively, relative to the prediction of GSM. Track errors in MSSG were larger than those of GSM. Adjustment of the surface flux scheme, which was done in the second stage to solve a precipitable water issue, may have reduced track errors in MSSG. Figure 3 addresses the time series of errors in TC track prediction in the second stage.hadhave better performance than diddoesGSMdependsimprovement ofconditionsimportant~~ required to improve track prediction.

We also validated MME using track predictions of the three models with reference to the RSMC Tokyo best-track data. MME track prediction gives  the smallest track errors for forecast time (FT) ofFT = 84 -120 h. The reduction rate of the MME position error from that of GSM was ~2624% at FT = 120 h relative to that of GSM. The position error of MME at that FT corresponds to that of GSM at FT = 96 h. 102 h.  promising results with regard to improving TC track prediction, future work is required to achieve more robust results and to answer scientific and practical questions, such as in which cases is MME effective and why

**5.3 Intensity predictions**

Figure 4 shows time series of the average central pressure and the standard deviation  in each model relative to the RSMC Tokyo best-track data for the  second stage. Because the global objective analysis data, which was used as initial conditions of the numerical experiments, tend to reproduce TC central pressure shallower than those in RSMC Tokyo best-track data, cases with an initial bias <20 hPa are validated. The central pressures in  MSSG and NICAM showed relatively small bias compared with the error in GSM. These results indicate that  7-km mesh models help decrease systematic positive errors for the central pressure. However, the central pressure in DFSM showed over-intensification and the magnitude of the bias after FT = 54 hours became larger

worth after  than that in GSM. At least, the comparison between GSM and DFSM  both DFSM and GSM those models ve the same specifications except for  horizontal resolution, this result suggests that the improvement  of  physics schemes suitable for such high-resolution models is needed for accurate forecasts of the central pressure.~~pressure TC intensity. This improvement is attributed to reduction of TC track forecast error (~100 km at FT = 120 h; Fig. 3) and better representation of TC structures, as shown in the next subsection.Theshowed abyhfirsthwhenintensityof the central pressure decrease by FT = 24 hthethis initial decreasebeganshowed ashowsofbyherror,keeps almostbyhshowsof negative bias forbygrowthhwhereashasbyh, Noticeably,showed a negativeh This demonstrates the advantage of MME for TC intensity and track prediction.~~

~~To evaluate characteristics of TC intensity prediction for each model, we show scatter diagrams of the relationship between predicted and RSMC Tokyo best-track central pressures (Fig. 5). First, GSM could not generally reproduce a central pressure lower than 940 hPa. DFSM sometimes reproduces a central pressure lower than 910 hPa, although such a TC frequently over-intensified after FT = 72 h. One of the reasons for such excessive intensification is the use of the same physical schemes tuned for a 20-km mesh model as that in the GSM. Through sensitivity experiments on the cumulus parameterization and cloud scheme, we confirmed that a modified physical scheme suitable to DFSM with 7 km horizontal resolution decreased the over-intensification (not shown). MSSG and NICAM reproduce a central pressure of nearly 930 hPa with relatively small standard deviation relative to the best-track data. From the standpoint of intensification rate, however, MSSG and NICAM still predicted rapid deepening of central pressure with difficulty, particularly the initiation of intensification.~~

**5.3 Predictions of the TC wind structure**

Accurate predictions of AR50 and AR30 lead to accurate estimations of the area affected by TCs. Figure 5 shows the validation result of AR50 based on the RSMC Tokyo best-track data. All models had negative bias of 80–90 km even at the initial time. This negative bias is partially is partly attributed to the shallower estimation of the central pressure by  ~5 hPa (Figure 4) associated with the biases in the underestimation of  global objective analysis data, which was used as initial conditions of the numerical experiments. TheThe difference in the interpolation methods to prepare the initial data for each model might also affect the bias. The negative biases of aall 7-km models reduced the initial negative bias  in the early stage. The negative bias of DFSM monotonically decreases until FT = 78 hours and then saturates at  ~negative bias of 25 km at FT = 78–120 hours. The bias of MSSG decreases more rapidly until FT = 48 hours and becomes positive until FT = 84 hours and then returns to a negative bias of ~20 km. The bias of NICAM continuously decreases until FT = 66 hours then becomes positive . At FT = 120 hours, NICAM shows a positive bias of 40 km, which was a smaller magnitude than that of the initial bias. Conversely, GSM shows little improvement in the negative bias so that its  negative bias remains still at of 60 km at FT = 120 hours. These results show that high-resolution models can significantly reduce the error of AR50


impact is expected. In addition, MME has a promising result in improving the AR50 prediction: MME showed thea bias became almostof nearly zero atforbias FT = 60–120 hhours.

Figure 6 shows the validation results of AR30. All models showed theshow a the-negative bias of more than 200 km at FT = 0 hhours. The negative biases of aAll 7-km models tended to decreasedecreasereduced the initial negative bias at the initial time in the early stage as FT proceeded. The negative bias of DFSM decreaseddecreasesreduced the negative bias bydecreases to 180 km byup toby FT = 36 h.hours and then relatively slowly decreaseddecreasesdecreasesreduced the bias byto 150 km byup toby FT = 120 hhours. The negative bias of MSSG shows temporally increasedtemporarily increases in the first 6 h,hourshnegative bias at FT = 0–6 h, and then decreaseddecreases.the negative bias d the negative bias. The bias of NICAM continuously decreaseddecreasesreduceddecreases the negative bias up to FT = 120 hhours, resulting in thea negative bias as small asasof 35 km at FT = 120 hhours. GSM hadsve little improvement onin AR30 up to FT = 96 hours and shows thea negative bias of about ~170 km at FT = 120 hours. and little improvement can be seen These results show that high-resolution models can also reduce the error ofin AR30. However, all the models still had relatively larger negative biases comparing with compared to the error ofin AR50. TowardTowards aThereforeToward better prediction of TC wind structure, further improvementimprovements in the quality of theofof objective analysis and the models themselvesanalysis with the output by the modelsmodel diagnosis should beare needed. The bias of MME also decreaseddecreasesdecreasesreduced the negative bias up to FT = 120 h, buthours; however, its magnitude wasis larger than that ofinof NICAM.

AccurateAn accurate prediction of the three-dimensional TC structure cancanmay lead to accurate predictions of the intensity, AR30 and AR50 prediction. Because there is no high-resolution TC observation whichthat is suitable for the validation of the simulated TC structure, here we made anaa intercomparison of the compare TC wind structures simulated by the 7-km models and 20-km mesh GSM. Figure 7 shows a composite of the radius-height section of the azimuthal mean radial and tangential wind speeds structure for TCs at the time of the RSMC Tokyo best-track analysed central pressure between 920–940 hPa, corresponding, in in the lifecycle, RSMC Tokyo best track data which ingsto the mature stage of a TC in the lifecycle. Total. A total of 347 snapshots were used for the composite analysis. If the models can perfectly simulate the TC structure perfectly, the result should be the same among thefor all models. WhereasWhile all 7-km mesh models reproduced a typical axisymmetric mean inner-core structurestructures, such as primary and secondary circulations, the simulated TC structures quite differed amongsignificantly between thediffers among 7-km models. The obviously shows diversity TCs calculated by in the DFSM had the highest maximum tangential wind speed and the smallest radius of maximum wind (RMW) amongof thethe7 km models. In addition, theitst primary circulation structure was the deepest which reaches, reaching up to 100 hPa in the vertical direction and wasis the narrowest in the horizontal direction. The depth of the inflow and outflow layers in DFSM was relatively thin amongof the models withand had the strongest radial velocity. The TCs in NICAM and MSSG showed have relatively similar structurestructures to each other, but; however, MSSG had thickerrelatively thickner inflow and /outflow layers. DifferenceDifferences in the heating and inertial stability in the inner-core lead to such differencedifferences in the primary and secondary circulation (Shapiro and Willoughby 1982). The differences between DFSM and these two models suggest different inner-core conditions in thermal and inertial stability (Shapiro and Willoughby 1982). Understanding the cause of differencethe differences in the simulated structures amongin the models will must lead to improvements in all the improvement of eacheachthe models.

Figure 6 shows horizontal distributions of hourly precipitation overlaid on SLP for Typhoon Wipha at 14 October 2013, 06:00:00 UTC (FT = 96 h) initiated at 10 October 2013, 06:00:00 UTC as an example. At that time, RSMC Tokyo best track data showed that central pressure, maximum wind speed, and radius of surface wind speeds of 25 m s$^{-1}$ (R25) were 940 hPa, 80 knot (40 m s$^{-1}$), and 120 nautical miles (220 km), respectively. Satellite observation (Fig. 7) suggests that convection in the inner core had an asymmetric structure and was most active in the northeastern semicircle, with spiral rain bands. GSM simulates a very weak TC (980 hPa), with maximum surface wind speed smaller than 25 m s$^{-1}$ and a weak and

disorganized precipitation pattern compared with those from DFSM, MSSG, and NICAM. DFSM has the most intense (897 hPa) and compact eyewall structure among the three 7 km mesh models, with R25 of ~95 km. In addition, the TC predicted by DFSM has double eyewalls. The difference of precipitation and SLP patterns between GSM and DFSM is attributed to their contrasting horizontal resolutions, because both models use the same configuration and specifications except for

5 horizontal resolution. MSSG and NICAM simulate intensities (934 and 953 hPa, respectively) and R25 values (approximately 265 and 175 km axisymmetric means, respectively) similar to RSMC best-track analyses. However, precipitation patterns are completely different. MSSG shows a concentric eyewall, represented by a well-organized circular precipitation pattern. The horizontal scale of the eyewall is wider than that of the DFSM. NICAM predicts a band-shape precipitation pattern, indicating that the simulated TC does not establish a concentric eyewall as in DFSM and MSSG. Even

10 though both MSSG and NICAM use explicit microphysical schemes without any cumulus parameterization, there are significant differences in the simulated TC structures.

Composite analyses of a radial-height section were done for Typhoon Wipha at the time of maximum intensity during its lifetime, using the results of 15 experiments (Table 1). Figure 8 shows radius height cross sections of azimuthal mean radial and tangential wind speeds. DFSM realistically reproduces the secondary circulation of a typical TC,

15 represented by inflow toward the TC centre in the lower troposphere and outflow in the upper troposphere, compared with the secondary circulation of the GSM. However, the axisymmetric structure predicted by DFSM differs greatly from that by MSSG and NICAM. The simulated inflow layer in MSSG is the thickest among the three models, more than double that of DFSM. Another unique structure from MSSG is inflow just below the upper outflow layer. The TC vortex height in NICAM is shallower than that of DFSM and MSSG. For example, the maximum height of tangential wind speed = 15 m s$^{-1}$ is ~100

20 hPa for DFSM and MSSG but 170 hPa for NICAM. The radius of maximum winds (RMW) in NICAM is more than twice that of DFSM. The slope of RMW simulated by NICAM is larger than that of DFSM and MSSG. Even though the horizontal resolution of the models is identical, differences in specifications such as dynamics and physical processes yields substantial differences in TC inner core structure.

Figure 9 shows the relationship between maximum axisymmetric mean tangential wind speed and RMW. X marks

25 show averages within bins every 5 m s$^{-1}$. The GSM is unable to reproduce the RMW derived from extended best-track data (mean or median RMWs are 64.6 and 55.5 km; Kimball and Mulekar, 2004), because the predicted RMW is > 100 km. Skamarock (2004) stated that seven times the horizontal grid spacing is the scale of the finest resolvable modes, which corresponds to ~140 km for GSM. Thus, it is difficult to reproduce an RMW < 100 km. The resolvable scale of the 7 km mesh model is ~50 km. MSSG and NICAM are able to reproduce the reduction in RMW with TC intensification, with a

30 mean simulated RMW > 50 km. The reduction in RMW is consistent with observation by aircraft penetration (e.g., Fig. 12 of Stern et al., 2015). The RMW predicted by DFSM is the smallest among the four models. We need sensitivity studies to clarify which factors cause the RMW differences, which are closely related to differences in vertical structure of the inner core (Fig. 8).

**6 Conclusions and future work**

35 The TYMIP-G7 project have had beenwas implemented in two stages, from June 2015 through March 2016. The aim of the project iswas to statistically quantify and understand the advantageadvantages of high-resolution, global atmospheric models toward the improvement ofto improve 5-day TC track, and 
[revised manuscript text omitted]

**Table 5 Computational performance in 12 September 2013, 00:00:00 UTC case**

| Model | Time step (s) | Number of nodes | Elapse time (sec) (including output of model data) | Node×hours | Execution efficiency (%) |
|---|---|---|---|---|---|
| DFSM | 200 | 320 | 7673 | 682 | 4.0 |
| MSSG | 17.7 | 512 | 16381 | 2330 | 15.1 |
| NICAM | 30 | 640 | 6497 | 1155 | 16.5 |
| GSM | 400 | 10 | 5896 | 16.4 | 16.0 |

**DFSM**
Reduced linear equally-spaced latitude grid

**NICAM**
Icosahedral grid

**MSSG**
Yin-Yang grid

**Figure 1:.** Schematic diagram of  horizontal grid structures of  three models used in TYMIP-G7

[Figure]

**Figure 2 Screen capture of the Web application: outgoing longwave radiation at 14 September 2013, 10:00:00 UTC simulated in experiments initialized at 12 September 2013, 06:00:00 UTC.**

[Figure]

文章校正を行う

[Figure]

Figure 3. Errors in the track prediction for GSM, DFSM, MSSG, NICAM and MME (in the second stage). Each grey bar indicates the number of samples at each forecast time (right–vertical axis). Error bars indicate 95% confidence levels of the central pressure difference between the prediction and the RSMC Tokyo best–track data.

[Figure]

[Figure]

[Figure]

**Figure 4.** Errors in the predictions of the central pressure for GSM, DFSM, MSSG, NICAM and MME (in the second stage). Each grey bar indicates the number of samples at each forecast time (right vertical axis). Error bars indicate 95% confidence levels of the central pressure difference between the prediction and the RSMC Tokyo best-track data.

[Figure]

[Figure]

**Figure 5.** Errors in the averaged radius of the 50--knot wind (AR50) for GSM, DFSM, MSSG, NICAM and MME (in the second stage). Each grey bar indicates the number of samples at each forecast time (right—vertical axis). Error bars indicate 95% confidence levels of the AR50 difference between the prediction and the RSMC Tokyo best--track data.

5

[Figure]

[Figure]

**Figure 6.** **Errors in the averaged radius of the 30--knot wind (AR30) for GSM, DFSM, MSSG, NICAM and MME (in the second stage). Each grey bar indicates the number of samples at each forecast time (right—vertical axis). Error bars indicate 95% confidence levels of the AR30 difference between the prediction and the RSMC Tokyo best--track data.**

5  ~~Horizontal distributions of precipitation (colour), sea level pressure (black contour) and wind speed 25 m·s⁻¹ (red contour) for Typhoon Wipha at FT = 96 h. Labels on horizontal and vertical axes shows zonal and meridional distances from TC centre (km), respectively. Contour intervals of sea level pressure are 10 hPa for > 960 hPa and 20 hPa for < 960 hPa. Plotted area is a 1000-km square and (x, y) = (0, 0) is set to TC centre.~~

[Figure]

**Figure 7. Composite analysis of height cross section of  axisymmetric mean radial (shaded) and tangential (contour) wind speed for TCs at the time of  analysed central pressure between 920940 hPa in  RSMC Tokyo best-track data. Contour intervals are 5 m s$^{-1}$ (values > 15 m s$^{-1}$ are plotted). The green line depicts the RMW between 850 hPa and 200 hPa. The grey shading at  bottom of each panel is below the surface.**

ントの色：赤

[Figure]

**Figure 7 Brightness temperature observed by Advanced Microwave Sounding Unit – A (AMSU-A) channel 89 GHz onboard NOAA-18 at 14 October, 2013 06:28 UTC. (image courtesy of Naval Research Laboratory). Red "X" displays TC Wipha centre.**

[Figure]

[Figure]

**Figure 8 Composite analysis of r-z cross sections of axisymmetric mean radial (shaded) and tangential (contour) wind speeds. Contour intervals are 5 m s⁻¹ (values > 15 m s⁻¹ are plotted). Green line depicts RMW between 850 and 200 hPa. Grey shading at bottom of each panel is below the surface.**

[Figure]

[Figure]

Figure 9 Scatter plot of maximum axisymmetric mean tangential wind speeds (x axis; m s⁻¹) and radius of maximum wind speeds (RMW, in km). Colours show forecast times (in h) of 3–24 (grey), 27–48 (red), 51–72 (green), 78–96 (blue), and 99–120 (cyan). X marks indicate mean RMW for each intensity (e.g., "20" uses all cases in which intensity is between 15 and 25 m s⁻¹)

---

## Author Response (AR2)

Reply to reviewer 1

The authors deeply appreciate your careful reading and the encouraging messages. We have corrected typos and all comments are mentioned in the revision. Point-to-point response are following.

*The authors have addressed the majority of my concerns regarding this manuscript. While the results contained are not exactly novel (e.g., increasing resolution and using multi-model ensembles improve TC forecasts), the paper should now be a sufficient reference for future studies using TYMIP-G7 data to address the many (interesting) questions that are raised in the last section of this manuscript. In particular, I look forward to seeing further studies that study the impact of model configuration on forecasted TCs and the relationship between maximum point-wise intensity and 3-D structure (there are many features worth exploring in Fig. 7 alone). I have relatively minor comments (primarily points of clarification and typos) that should be addressed, but otherwise I feel the manuscript is suitable for publication in GMD.*

Thank you for your careful reading and the encouraging comments.

*Page 1, Line 33; "furnish" is awkward. Replace with "produce."*

Corrected. Thanks.

*Page 2, Line 33; "Errors in TC track prediction by the JMA operational global atmospheric model at a given lead time have..."*

Corrected. Thanks.

*Page 2, Line 39; "... Fenshen predicted recurvature far from the..."*

Corrected. Thanks.

*Page 4, Line 8, "calculated" should be "forecasted"*

Corrected. Thanks.

*Page 6, Line 11, "MSSG-A is primarily used..." this seems exclusively used (I don't see any coupling for MSSG-O in this paper). It should be noted if any simulations do indeed couple to MSSG-O.*

Corrected as "MSSG-A is exclusively used⋯." Thank you for the comment.

*Page 8, Lines 5-6, How is dissipation determined? When a local SLP no longer exists?*

We determined the dissipation when difference between the minimum SLP and an ambient SLP defined as an areal average within 500 km of the minimum SLP point is less than 1 hPa. We have described this point (Page 8, Lines 5-6).

*Page 10, Lines 1-2. It is interesting that NICAM tends to have the largest AR50s (indicating a broad wind field) but DFSM has a much larger intensity bias (towards stronger storms). This implies that the internal dynamics of the modeled TCs are very different between the two simulations (DFSM has a very tight inner core versus NICAM which has a weaker, broader core, but also a broader wind field).*
*Interestingly, this actually allows for prediction of what Fig. 7 will look like before it is introduced. The authors could note this specific point as an area that needs to be rigorously explored to understand why the two models produce vastly different TCs (I assume due to both dynamical core and physical parameterizations).*

Thank you for the comment. We strongly agree with you and have summarized this interesting points and mentioned future direction of study to understand the diversity of modelled TC (Page 12, Lines 12-17).

*Page 24, Line 24, should be "atmosphere-only"*

Corrected (Page 11). Thanks.

*Page 24, Line 26, another relevant reference may be Zarzycki (2016, Tropical cyclone intensity errors associated with lack of two-way*

*ocean coupling in high-resolution global simulations) who showed that a simple slab ocean was not sufficient to properly represent TC cold wakes and that a turbulent mixing parameterization is required.*

We cited Zarzycki (2016) in P11, Line 32-35. Thank you for the suggestion.

*Page 24, Line 27, "aa" typo.*

Corrected. Thanks.

[revised manuscript text omitted]
. Interestingly, DFSM tended to simulate the small wind radii (AR50 and AR30) despite of the largest negative bias for central pressure. NICAM and MSSG, which had smaller biases for central pressure, tended to simulate larger wind radii than DFSM. Therefore, it is expected that simulated TCs in NICAM and MSSG have horizontally broader structure than that in DFSM. These results imply that internal dynamics of modelled TC are significantly different among those models. Further studies are needed to understand the differences in internal dynamics of modelled TC by changing physics parameterization and dynamical core.

[revised manuscript text omitted]

Japan Meteorological Agency: Outline of the operational numerical weather prediction at the Japan Meteorological Agency. Appendix to WMO technical progress report on the global data-processing and forecasting system and numerical weather prediction,188p.http://www.jma.go.jp/jma/jma-eng/jma-center/nwp/outline2013-nwp/index.htm, 2013.

Japan Meteorological Agency: Annual rReport on the aActivities of the RSMC Tokyo -tTyphoon cCenter, 90p, http://www.jma.go.jp/jma/jma-eng/jma-center/rsmc-hp-pub-eg/AnnualReport/2014/Text/Text2014.pdf, 2014.

Japan Meteorological Agency: The uUpgrade hHistory of the gGlobal sSpectral mModel. http://www.wis-jma.go.jp/ddb/latest_modelupgrade.txt, 2016.

Joint Typhoon Warning Center: 2008 Annual tTropical cCyclone rReport, 116p, http://www.usno.navy.mil/NOOC/nmfc-ph/RSS/jtwc/atcr/2008atcr.pdf

Kageyama, A. and Sato, T.: The Yin-Yang gGrid: An oOverset gGrid in sSpherical gGeometry. Geochem. Geophys.Geosyst., 5, Q09005, doi:10.1029/2004GC000734, 2004.

[revised manuscript text omitted]

Takata, K, Emori, S., and Watanabe, T.: Development of the minimal advanced treatments of surface interaction and runoff. Global and Planet. Change, 38, 209-222, 2003.

Taniguchi, H., Yanase, W., and Satoh, M.: Ensemble sSimulation of cCyclone Nargis by a gGlobal cCloud-sSystem-rResolving mModel—mModulation of cCyclogenesis by the Madden-Julian oOscillation, J. Meteor. Soc. Jpn., 88, 571-591, doi: 10.2151/jmsj.2010-317, 2010.

Tomita, H., Tsugawa, M., Satoh, M., and Goto, K.: Shallow wWater mModel on a mModified iIcosahedral gGeodesic gGrid by uUsing sSpring dDynamics, J. Comp. Phys., 174, 579-613, doi:10.1006/jcph.2001.6897, 2001

Tomita, H.: New microphysical schemes with five and six categories by diagnostic generation of cloud ice, J. Meteorol. Soc. Japan, 86A, 121-142. doi: 10.2151/jmsj.86A.121, 2008.

Wang, B., and Rui, H.: Synoptic climatology of transient tropical intraseasonal convection anomalies: 1975–1985, Meteorol. Atmos. Phys., 44, 43–61, 1990.

Wang, B., and Xie, X.: A model for the boreal summer intraseasonal oscillation, J. Atmos. Sci., 54, 72–86, 1997.

Wang, H., and Wang, Y.: A numerical study of typhoon Megi (2010): Part I: Rapid intensification. Mon. Wea. Rev., 142, 29-48, doi:10.1175/MWR-D-13-00070.1, 2014.

Wedi, N. P. and Smolarkiewicz, P. K.: A framework for testing global non-hydrostatic models. Q. J. R. Meteor. Soc., 135, 469-484, doi: 10.1002/qj.377, 2009.

Wicker, L. J. and Skamarock, W.C.: Time-split-ting methods for elastic models using forward time schemes, Monthly Weather Review, 130, 2088–2097, 2002.

Xiang, B., Lin, S.-J., Zhao, M., Zhang, S., Vecchi, G., Li, T., Jiang, X., Harris, L., and Chen, J.-H.: Beyond weather time-scale prediction for hurricane Sandy and super typhoon Haiyan in a global climate model. Mon. Wea. Rev., 143, 524-535, doi: 10.1175/MWR-D-14-00227.1, 2015.

Yablonsky, R. M., and Gienis, I.: Limitation of oOne-dDimensional oOcean mModels for cCoupled hHurricane–oOcean mModel fForecasts, Mon. Wea. Rev., 137, 4410–4419, doi: 10.1175/2009MWR2863.1, 2009.

Yabu, S.: Development of longwave radiation scheme with consideration of scattering by clouds in JMA global model, CAS/JSC WGNE Research Activities in Atmospheric and Oceanic Modelling. 43. 4.07-4.08, 2013.

Yamada H, Nasuno T, Yanase W, and Satoh M: Role of the vertical structure of a simulated tropical cyclone in its motion: A case study of Typhoon Fengshen (2008), SOLA, 12, 203–208, doi:10.2151/sola.2016-041041, 2016.

Yamaguchi, M., Iriguchi, T., Nakazawa, T., and Wu, C.-C.: An observing system experiment for Typhoon Conson (2004) using a singular vector method and DOTSTAR data. Mon. Wea. Rev., 137, 2801-2816, 2009.

Yamaguchi, M., Nakazawa, T. and Hoshino, S.: On the relative benefits of a multi-centre grand ensemble for tropical cyclone track prediction in the western North Pacific. Quart. J. Roy. Meteor. Soc., 138, 2019–2029, doi:10.1002/qj.1937, 2012.

Yamaguchi, M., Vitart, F., Lang, S. T. K., Magnusson, L., Elsberry, R. L., Elliott, G., Kyouda, M., and Nakazawa, T.: Global distribution on the skill of tropical cyclone activity forecasts from short- to medium-range time scales. Wea. Forecast, 30, 1695-1709, doi: 10.1175/WAF-D-14-00136.1, 2015.

Yanase, W., Taniguchi, H., and Satoh, M.: The genesis of tropical cyclone Nargis (2008): Environmental modulation and numerical predictability. J. Meteor. Soc. Japan, 88, 497-519, doi: 10.2151/jmsj.2010-314, 2010.

Yoshimura, H. and Matsumura, T.: A Semi-Lagrangian sScheme cConservative in the vVertical dDirection. CAS/JSC WGNE Research Activities in Atmospheric and Ocean Modeling, 33, 3.19-3.20, 2003.

Yoshimura, H. and Matsumura, T.: A two-time-level vertically-conservative semi-Lagrangian semi-implicit double Fourier series AGCM. CAS/JSC WGNE Research Activities in Atmospheric and Ocean Modeling, 35, 3.25-3.26, 2005.

Yoshimura, H.: Development of a nonhydrostatic global spectral atmospheric model using double Fourier series. CAS/JSC WGNE Research Activities in Atmospheric and Ocean Modeling, 42, 3.05-3.06, 2012.

Yukimoto, S., and Coauthors: Meteorological Research Institute-Earth System Model Version 1 (MRI-ESM1)–Model Description–. Technical Reports of the Meteorological Research Institute, No. 64, 2011.

Zarzycki, C. M.: Tropical cCyclone iIntensity eErrors aAssociated with lLack of tTwo-wWay oOcean cCoupling in hHigh-rResolution gGlobal sSimulations. J. Clim., 29, 8589–8610, doi: 10.1175/JCLI-D-16-0273.1.

[revised manuscript text omitted]